# Dendritic coincidence detection in Purkinje neurons of awake mice

Christopher J Roome*, Bernd Kuhn*

Optical Neuroimaging Unit, Okinawa Institute of Science and Technology Graduate University (OIST), Okinawa, Japan

**Abstract** Dendritic coincidence detection is fundamental to neuronal processing yet remains largely unexplored in awake animals. Specifically, the underlying dendritic voltage–calcium relationship has not been directly addressed. Here, using simultaneous voltage and calcium two-photon imaging of Purkinje neuron spiny dendrites, we show how coincident synaptic inputs and resulting dendritic spikes modulate dendritic calcium signaling during sensory stimulation in awake mice. Sensory stimulation increased the rate of postsynaptic potentials and dendritic calcium spikes evoked by climbing fiber and parallel fiber synaptic input. These inputs are integrated in a time-dependent and nonlinear fashion to enhance the sensory-evoked dendritic calcium signal. Intrinsic supralinear dendritic mechanisms, including voltage-gated calcium channels and metabotropic glutamate receptors, are recruited cooperatively to expand the dynamic range of sensory-evoked dendritic calcium signals. This establishes how dendrites can use multiple interplaying mechanisms to perform coincidence detection, as a fundamental and ongoing feature of dendritic integration in behaving animals.

## Introduction

Dendritic integration is fundamental to signal processing in the brain. So far, most studies on dendritic integration have been performed in vitro, in the absence of physiological inputs (*Larkum et al., 2009*; *Markram et al., 1997*; *Stuart and Häusser, 2001*; *Wang et al., 2000*). Equivalent in vivo studies, though technically challenging, are essential for exploring how dendritic integration works in living animals (*Chen et al., 2013*; *Jia et al., 2010*; *Murayama et al., 2009*; *Palmer et al., 2014*; *Sheffield and Dombeck, 2015*; *Smith et al., 2013*).

Conventional in vivo dendritic recording techniques have faced important challenges for studying dendritic integration. Notably, in vivo patch clamp recordings targeting the soma or smooth dendrites cannot directly measure signals in the most distal dendritic regions that receive the majority of synaptic input, called the 'spiny' dendrites. In vivo calcium imaging from spiny dendrites can partially circumvent this issue (*Chen et al., 2012*). However, this approach measures dendritic calcium events only and omits the depolarizing and hyperpolarizing voltage signals evoked by synaptic input that do not trigger postsynaptic calcium signals. These synaptic potentials are expected to generate continuous but highly inhomogeneous spatiotemporal dendritic activity in awake animals. A glimpse of this was shown in the cerebellar Purkinje neuron (PN) (*Roome and Kuhn, 2018*). As such, our understanding of the basic components of dendritic integration, the frequency, amplitude, and spatiotemporal distribution of synaptic inputs under physiological conditions, and how these inputs are integrated by dendrites in awake behaving animals, remains incomplete.

Coincidence detection is a basic form of dendritic integration. By detecting coincident synaptic input, it is thought that neurons distinguish important signals from ongoing synaptic activity by amplifying dendritic voltage signals as part of ongoing synaptic integration (*Stuart and Häusser, 2001*) and by modifying synaptic strength through synaptic plasticity (*Brown et al., 1990*). PN dendrites are ideally suited to perform coincidence detection. They receive excitatory synaptic input

**\*For correspondence:**
christopher.roome@oist.jp (CJR);
bkuhn@oist.jp (BK)

**Competing interests:** The authors declare that no competing interests exist.

from two distinct pathways: a single climbing fiber (CF) and numerous parallel fibers (PFs). CFs project from the inferior olive and evoke reliable dendritic calcium spikes. PFs relay mossy fiber (MF) activity originating from multiple brainstem nuclei, and together with synaptic input from inhibitory molecular layer interneurons (MLIs) elicit postsynaptic potentials in the PN dendrites (*Kitamura and Häusser, 2011*; *Konnerth et al., 1990*; *Roome and Kuhn, 2018*). In cerebellar slices, paired stimulation of PF and CF input evoke 'supralinear' dendritic calcium signals in PNs, whereby the signal amplitude is larger than the sum of calcium signals triggered by PF and CF input alone. At PF–PN synapses these supralinear calcium signals lead to a form of synaptic plasticity, known as long-term depression (LTD) (*Miyata et al., 2000*; *Wang et al., 2000*).

This particular coincidence detection process is well studied in PN spines and dendrites in cerebellar slices, and supports early theoretical predictions for motor learning in the cerebellum whereby the CF input is thought to serve as an instructive 'teaching' signal that modifies PF synaptic input through induction of LTD (*Marr, 1969*). However, this remains a controversial component in the theory of cerebellar function and motor learning, and a corresponding description of dendritic coincidence detection from awake animals is still missing (*Mauk et al., 1998*; *Najafi and Medina, 2013*; *Sakurai, 1987*; *Schonewille et al., 2011*; *Wang et al., 2000*). Specifically, previous in vitro studies identified two dendritic mechanisms by which coincidence detection may occur, which depend on the strength and timing of coincident PF and CF stimulation (*Wang et al., 2000*). These mechanisms involved voltage-gated calcium channels (VGCCs) and/or group one metabotropic glutamate receptors (mGluR1). How these processes occur during behavior remains to be determined.

In vivo studies using calcium imaging found that CF-evoked calcium signals were enhanced during sensory stimulation (an air puff directed toward the eye) in mice. Notably, graded non-CF sensory-evoked calcium signals were detected and proposed to be driven by PF input (*Najafi et al., 2014a*; *Najafi et al., 2014b*). More recent studies, however, have failed to detect supralinear dendritic calcium signals triggered by coincident PF and CF input during sensory stimulation (*Gaffield et al., 2019*; *Gaffield et al., 2018*), challenging the role that PF input might play on the sensory-evoked dendritic calcium signals.

Coincidence detection of PF and CF input remains to be confirmed in vivo. Specifically, it is unclear if the coincidence detection mechanisms previously described in vitro also occur in vivo and if so, how they occur and under what behavioral conditions. Furthermore, because of technical limitations, a direct relationship between voltage and calcium signaling in spiny dendrites has not yet been explored in behaving animals.

Here by combining simultaneous voltage and calcium imaging from the PN spiny dendrites, we record dendritic calcium spikes and postsynaptic potentials to investigate sensory-evoked dendritic integration and coincidence detection in PN dendrites of awake mice. Specifically, we investigate the dendritic mechanisms that lead to the enhanced dendritic calcium signals in PNs during sensory stimulation.

## Results

To study dendritic integration in vivo, we recently developed a fast (2 kHz) two-photon imaging technique to simultaneously record dendritic voltage and calcium signals from PN spiny dendrites in awake mice. By recording dendritic voltage and calcium signals simultaneously we recorded fast dendritic calcium spikes and also slow depolarizing or hyperpolarizing synaptic inputs, which were confirmed using combined electrophysiology and pharmacological manipulation (*Kuhn and Roome, 2019*; *Roome and Kuhn, 2018*; *Roome and Kuhn, 2019*; *Roome and Kuhn, 2020*).

First, we will summarize the techniques we used for this study, and to connect to previous work done in vitro (*Wang et al., 2000*), we begin by describing electrically evoked dendritic signals in vivo, before exploring spontaneous and sensory-evoked dendritic signals.

### Electrical stimulation of PF input evokes graded dendritic voltage and calcium signals and dendritic spikes (DSs) in PN dendrites

Single PNs in lobule V of the cerebellar vermis were double labeled with voltage sensitive dye ANNINE-6plus and genetically encoded calcium indicator GCaMP6f, as described previously (*Roome and Kuhn, 2018*) (see Supplementary information). Briefly, a chronic cranial window with access port (*Roome and Kuhn, 2014*) permitted two photon imaging, single neuron labeling,

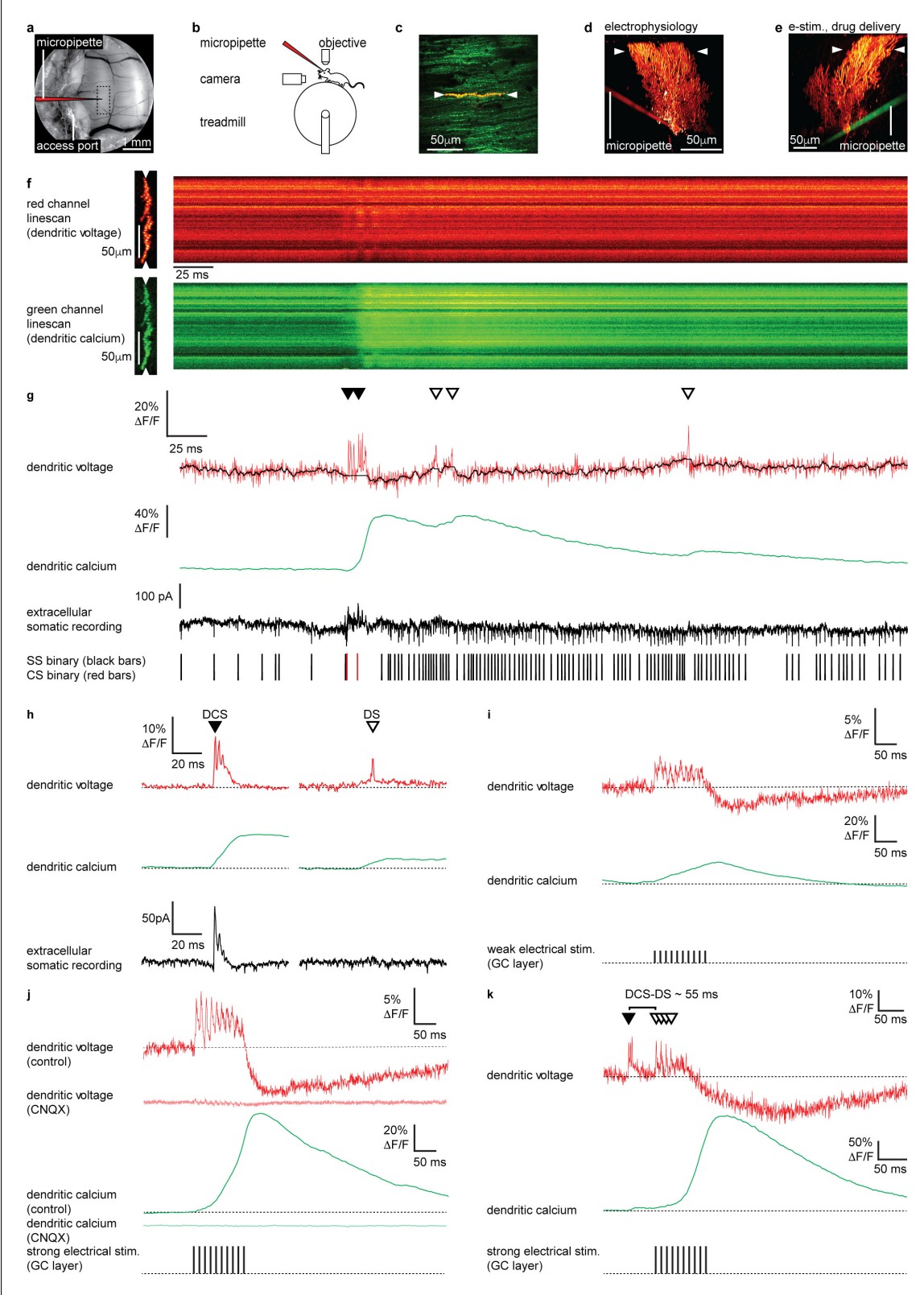

**Figure 1.** Electrical stimulation of parallel fiber (PF) input evokes graded dendritic voltage and calcium signals and dendritic spikes (DSs) in Purkinje neuron (PN) dendrites. (a) A chronic cranial window with access port on the cerebellar vermis lobule V was used for imaging from the dendrites of single PNs and allowed access to the brain via micropipette. The dashed box indicates the typical region used for PN labeling and recording (schematically indicated). (b) Sketch of the setup with a mouse mounted on a treadmill under a 2P microscope for imaging in the cerebellar vermis. (c)

*Figure 1 continued on next page*

*Figure 1 continued*

2P image of the spiny dendrites of a single PN labeled with voltage sensitive dye ANNINE-6plus (red) and genetically encoded calcium indicator GCaMP6f (green), resulting in double labeling (yellow), showing position of 2P linescan (white arrows). (d) Reconstruction of a single labeled PN showing position of 2P linescan (white arrows) and the micropipette (red) positioned on the PN soma used for electrophysiology. (e) Reconstruction of a different PN showing position of 2P linescan (white arrows) and the micropipette (green) positioned in the granular layer (approx. 50 µm below and lateral to the PN soma) used for electrical stimulation and pharmacological manipulation. (f) Single linescan recordings of simultaneous dendritic voltage (red channel) and calcium (green channel) from the PN shown in (d). (g) Spatial average of linescan recordings in (f) showing dendritic voltage (red trace) and calcium (green trace); black overlay shows voltage trace, with spikelets removed and 10 ms boxcar filtering. Simultaneous extracellular somatic recording (black trace) and corresponding binary sequence below show somatic simple spikes (SS; black bars) and complex spikes (CS; red bars). Dendritic calcium spikes are clearly visible in both voltage and calcium traces. Dendritic complex spikes (DCSs) are indicated by filled black triangles and dendritic spikes (DSs) are indicated by open triangles. (h) Averages of 32 DCS signals (left) and 25 DS signals (right), from 7 PNs, showing dendritic voltage (red), dendritic calcium (green), and extracellular somatic recording (black). (i) Average PN dendritic voltage (red) and calcium (green) in response to weak electrical stimulation (10 negative current pulses, 30 mA, 1 ms duration, at 100 Hz, schematically indicated) in the granule cell (GC) layer (26 averages, 7 PNs). (j) Average PN dendritic voltage (red) and calcium (green) in response to strong electrical stimulation (10 negative current pulses, 300 mA, 1 ms duration, at 100 Hz, schematically indicated) in the GC layer (26 averages, 7 PNs). Faint red and green traces show voltage (red) and calcium (green) in response to strong electrical stimulation, after CNQX (100 µM) application (15 averages, 1 PN). (k) An example of how PF input (here evoked by strong electrical stimulation in the GC layer) following a spontaneous climbing fiber (CF) event (DCS) can generate additional DS events and result in a supralinear dendritic calcium signal. The time interval between the spontaneous DCS and first evoked DS (DCS–DS) is indicated. Note the different scale bars compared to (i and j).

The online version of this article includes the following figure supplement(s) for figure 1:

**Figure supplement 1.** Micropipette injection into cerebellar granule cell (GC) layer guided by two-photon microscopy.

electrophysiology and drug delivery (*Figure 1a–e*, *Figure 1—figure supplement 1*). Extracellular recording was performed by positioning a recording micropipette on the PN soma (*Figure 1d*). Drug delivery and electrical stimulation of granule cells (GCs) were performed by positioning the micropipette in the GC layer, 50–100 µm below the PN layer and 50–100 µm lateral to the labeled PN. The micropipette was carefully placed lateral to the labeled PN to prevent damage and stimulation of the CF, which was confirmed by imaging dendritic voltage and calcium activity (*Figure 1e*).

Two photon linescan imaging (2 kHz) recorded dendritic voltage and calcium signals simultaneously from the spiny PN dendrites, predominantly comprising PF–PN synapses, at a depth of 30–70 µm below the dura (*Figure 1f*). Simultaneously acquired voltage and calcium linescans were spatially averaged to maximize the signal-to-noise ratio, at a temporal resolution of 0.5 ms/line. To maintain high temporal precision, voltage signals were not temporally filtered (unless stated otherwise), while all calcium signals (which are much slower) were temporally filtered (5 ms boxcar) to achieve an accurate measurement of the peak ΔF/F in single trials (*Figure 1g*). As the voltage response of ANNINE-6plus is linear down to a nanosecond time scale (*Kuhn and Roome, 2019*; *Roome and Kuhn, 2019*), the averaged voltage imaging signal represents the average membrane potential of dendritic shafts and dendritic spines within the point spread function of excitation along the scan line. It is important to keep in mind that spatially averaged recordings show the average of voltage and calcium signals across many dendritic processes, which we have previously shown are variable across individual branches (*Roome and Kuhn, 2018*), and so these signals should be viewed as an averaged dendritic signal across the width of the PN spiny dendrites. The relative fluorescence change of ANNINE-6plus can be used to estimate the average membrane voltage change in PNs with a conversion factor of 2.1 mV/% (*Kuhn et al., 2004*; *Kuhn and Roome, 2019*; *Roome and Kuhn, 2018*).

We previously described two types of dendritic spiking events in the PN dendrites of awake mice; dendritic complex spikes 'DCSs' triggered by CF input and single dendritic spikes 'DSs' that we proposed were triggered by strong PF input (*Roome and Kuhn, 2018*; *Figure 1g and h*) (see Materials and methods for DCS and DS detection criteria). DCS comprises a burst of rapid and distinct spikelets (typically two to five spikelets, each 1–2 ms in duration), associated with a large dendritic calcium signal. DS on the other hand are characterized by a gradually ramping membrane potential, followed by a single spikelet and small associated calcium signal. At the PN soma DCS are always associated with a somatic complex spike, while DS have no correlated somatic signal (compare extracellular somatic recordings in *Figure 1h*). We also described non-spiking dendritic voltage signals. These were either depolarizing or hyperpolarizing and correlated with action potential firing at the PN soma. All signals were blocked by using the AMPA receptor antagonist CNQX or sodium

channel antagonist Lidocaine, confirming their synaptic origin (*Roome and Kuhn, 2018*) (see also *Figure 2—figure supplements 7* and *8*).

We first established the dendritic voltage signals associated with PF synaptic input. Stimulation of PFs evokes graded excitatory postsynaptic potentials (EPSPs) and calcium signals in PN dendrites in vitro (*Eilers et al., 1995*). To examine these signals in vivo we electrically stimulated GCs in the GC layer, whose axons form the PFs, and recorded evoked voltage and calcium signals in the PN dendrites (*Figure 1i–k*). Physiologically plausible electrical stimulation protocols, similar to those used previously in vitro (*Wang et al., 2000*) (10 negative current stimuli, 30 mA or 300 mA, 1 ms duration at 100 Hz), were delivered to evoke PF synaptic input onto PN dendrites (*Figure 1i and j*).

Weak electrical stimulation (30 mA) evoked a depolarizing dendritic voltage signal (average during stimulus; 1.8 ± 2.1% ΔF/F, mean ± SD, n = 26, 7 PNs), corresponding to a sustained depolarization in membrane potential of approximately 3.8 ± 4.2 mV, and a gradual increase in dendritic calcium (calcium peak; 18.3 ± 23.0% ΔF/F, time to peak; 92.3 ± 46.9 ms, mean ± SD, n = 26, 7 PNs), and relatively few DS events (3 ± 2 DS/100 ms, mean ± SD, n = 26, 7 PNs) (*Figure 1i*). Strong electrical stimulation (300 mA) evoked a larger sustained depolarization (average during stimulus, 3.3 ± 2.0% ΔF/F, mean ± SD, n = 26, 7 PNs, unpaired t-test, p=0.0148), corresponding to a depolarization in membrane potential of approximately 6.9 ± 4.2 mV, and a larger supralinear dendritic calcium signal with a delayed time to peak (calcium peak; 75.2 ± 55.0% ΔF/F, unpaired t-test, p=1.1 × 10$^{-5}$, time to peak; 129.1 ± 26.1 ms, unpaired t-test, p=9.8 × 10$^{-4}$, mean ± SD, n = 26, 7 PNs). Strong electrical stimulation also triggered DS events more consistently (5 ± 3 DS/100 ms, mean ± SD, n = 26, 7 PNs, Kolmogorov–Smirnov, p=0.0308) (see *Figure 1j and k*).

A prolonged hyperpolarizing voltage signal also followed the depolarization that increased with stimulus strength (average 100 ms following weak stimulus; −1.7 ± 2.7% ΔF/F, average 100 ms following strong stimulus; −4.7 ± 5.1% ΔF/F, mean ± SD, n = 26, 7 PNs, unpaired t-test, p=0.01), which we propose is due to feed forward inhibition that follow PF-evoked depolarization and/or intrinsic dendritic mechanisms, such as voltage- or calcium-activated potassium channels, as described in vitro (*Martina et al., 2003*; *Mittmann et al., 2005*; *Otsu et al., 2014*; *Womack, 2004*; *Zagha et al., 2010*). AMPA receptors are expressed throughout the cerebellar circuitry, including PF–PN, CF–PN, PF–MLI, and MF–GC synapses (*Konnerth et al., 1990*; *Perkel et al., 1990*; *Yamazaki et al., 2010*). Accordingly, the depolarizing and hyperpolarizing voltage signals, DCSs, DSs and calcium signals were all blocked following application of CNQX (100 μM) confirming that these signals arise from EPSPs and inhibitory postsynaptic potentials (IPSPs) (see faint red and green traces in *Figure 1j*).

Taken together this indicates that dendritic calcium signals evoked by PF synaptic input have multiple calcium sources, as described previously in vitro (*Denk et al., 1995*; *Eilers et al., 1995*; *Llano et al., 1991a*; *Sugimori and Llinás, 1990*; *Wang et al., 2000*). These are likely to include calcium influx gated by mGluR1 activation at PF–PN synapses (*Tempia et al., 2001*), voltage-dependent DSs (*Denk et al., 1995*; *Roome and Kuhn, 2018*; *Usowicz et al., 1992*), and calcium released from internal stores (*Llano et al., 1994*; *Llano et al., 1991a*; *Wang et al., 2000*). Since PF and CF synaptic input constitute the only excitatory input to PN dendrites (*Llano et al., 1991b*), and DS events are blocked by CNQX (*Figure 1j*; see also *Figure 2—figure supplement 7*), have no corresponding somatic complex spike associated with CF input (*Figure 1h*), and are evoked by strong PF stimulation (*Figure 1i–k*), we conclude that the DS events are generated by strong PF input to the PN dendrites. As a proof of principle for dendritic coincidence detection of PF and CF input, *Figure 1k* shows an example of how PF input (here evoked by electrical stimulation) following a spontaneous CF event triggers additional DS events and results in a supralinear dendritic calcium signal. While electrical stimulation of GCs may be unphysiological here, this demonstrates how supralinear dendritic calcium signals can be generated by coincident PF and CF input to PN dendrites in vivo.

## Sensory stimulation evokes graded depolarizing and hyperpolarizing signals and calcium signals in PN dendrites

We next explored dendritic signaling evoked by sensory stimulation in awake mice. In total 54 PNs in 23 mice were labeled and several PNs (up to 5) were labeled in the same mouse (*Figure 2a*). Two photon linescan imaging (2 kHz) recorded dendritic voltage and calcium signals simultaneously from the spiny dendrites of labeled PNs. Throughout the experiment, fully awake mice (i.e., awake for at least 1 hr after isoflurane anesthesia; *Figure 2—figure supplement 5*) were seated on a rotating

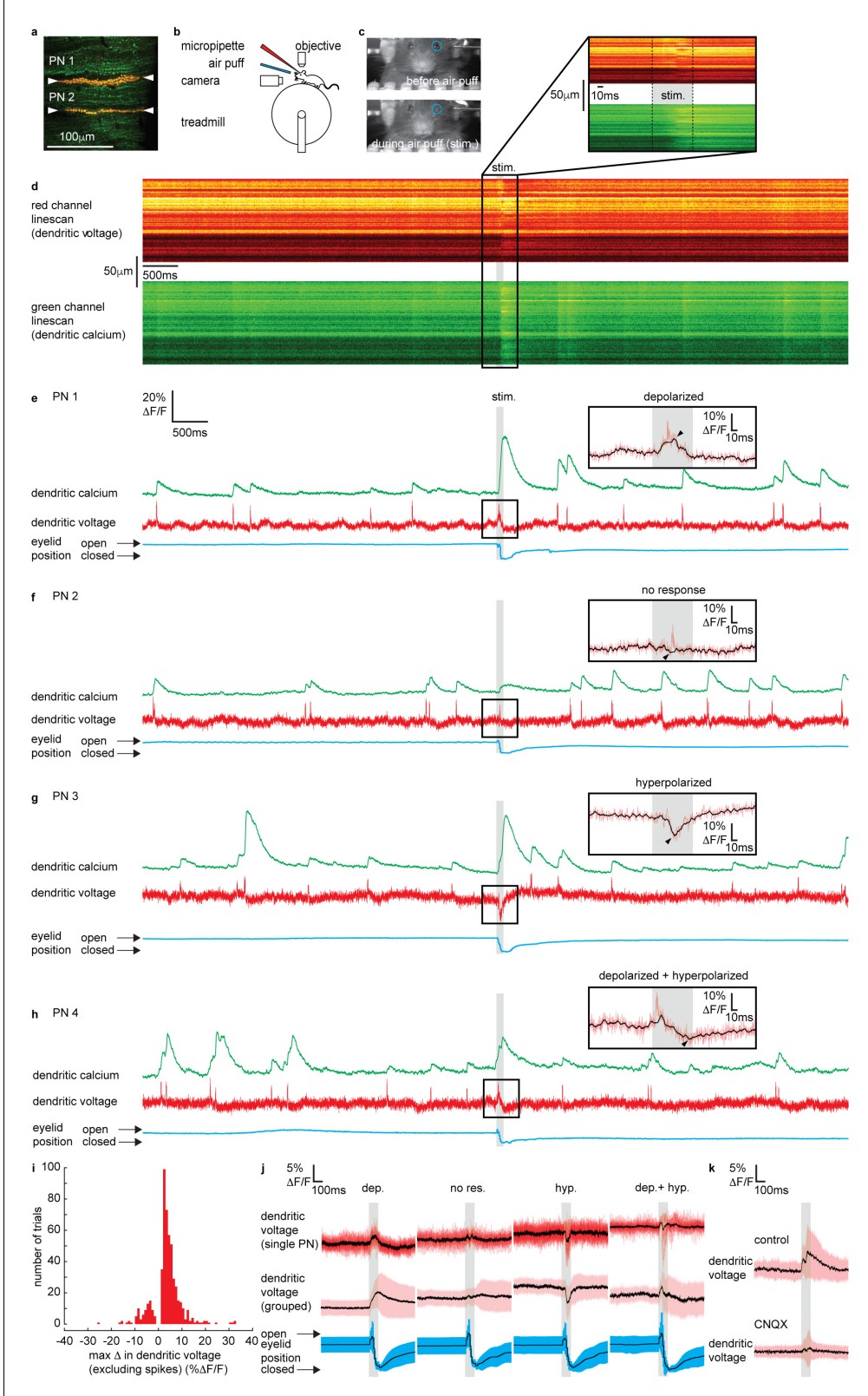

**Figure 2.** Sensory stimulation evokes graded depolarizing and hyperpolarizing signals and calcium signals in Purkinje neuron (PN) dendrites. (a) 2P image of the spiny dendrites of two PNs (PN1 and PN2) labeled with voltage sensitive dye ANNINE-6plus (red) and genetically encoded calcium indicator GCaMP6f (green), resulting in double labeling (yellow). (b) Sketch of the setup with a mouse mounted on a treadmill under a 2P microscope for imaging in the cerebellar vermis. A micropipette was used for drug delivery. A glass capillary was used to deliver a 100 ms air puff directed toward

*Figure 2 continued on next page*

*Figure 2 continued*

the ipsilateral eye. (**c**) A camera was used to monitor mouse movements and record eye responses during the air puff. Blue circles show regions of interest (ROIs) used to record eyelid movement. (**d**) Single linescan recordings of simultaneous dendritic voltage (red channel) and calcium (green channel) from the PN1 shown in (a). (**e–h**) Single trial recordings of simultaneous dendritic voltage (red traces) and calcium (green traces). DCS and DS events are visible in both voltage and calcium traces. Blue traces show average intensity of ROI used to record eye responses. Gray bars show full duration of air puff stimulus. Note that PN1 (**e**) and PN2 (**f**) recordings are from two PNs in the same mouse, shown in (a). The calcium recording from PN1 (e) shows a dramatically enhanced evoked calcium signal during the air puff while the calcium recording from PN2 (f) does not. Insets show zoomed voltage traces during the sensory stimulus. The voltage recording in PN1 inset shows depolarizing voltage signal, during the air puff (small black arrows indicate max Δ in dendritic voltage, and black trace; 10 ms boxcar filtered voltage recording), while the voltage recording from PN2 shows no underlying depolarizing signal. (**i**) Histogram for max Δ in dendritic voltage during the sensory stimulus using 1% ΔF/F bins for all recording trials, n = 696, 54 PNs. (**j**) Dendritic voltage signals during the sensory stimulus sorted into response groups, showing single trial recordings from a single PN (top traces; black trace shows mean, colored traces show single trials) and averages from all recordings in each group (middle traces; black trace shows mean, colored trace shows SD) and corresponding eyelid responses (bottom traces; black trace shows mean, colored trace shows SD). (**k**) Dendritic voltage signals during the sensory stimulus under control conditions (top; black trace shows mean, colored trace shows SD, n = 46 trials, 5 PNs) and after CNQX application (bottom; black trace shows mean, colored trace shows SD, n = 37 trials, 5 PNs).

The online version of this article includes the following figure supplement(s) for figure 2:

**Figure supplement 1.** No difference in dendritic voltage signaling or time to eyelid closure between first and last recordings.
**Figure supplement 2.** Dendritic voltage signals are time locked to sensory stimuli and not voluntary eye blinks.
**Figure supplement 3.** Example linescan showing evoked hyperpolarizing voltage signal during a sensory stimulus.
**Figure supplement 4.** Sensory stimulus-evoked dendritic voltage signals and their relationship with eyelid closure time and climbing fiber (CF)-evoked dendritic calcium signals.
**Figure supplement 5.** Isoflurane induced anesthesia blocks stimulus-evoked dendritic voltage and calcium signaling.
**Figure supplement 6.** During sensory stimulation the average maximum change in dendritic voltage in our sampling of lateral lobule V is positive and dendritic calcium signals are enhanced.
**Figure supplement 7.** AMPA receptor antagonist, CNQX, blocks stimulus-evoked dendritic voltage and calcium signaling.
**Figure supplement 8.** Na$^+$ channel antagonist, Lidocaine, blocks stimulus-evoked dendritic voltage and calcium signaling and slows eyelid closure time.

treadmill and head-fixed under a two-photon microscope. Linescan recordings (10 s in length) included a 100 ms sensory stimulus (stim.) beginning 5 s after recording onset, delivering an air puff directed at the ipsilateral eye. High intensity air puffs (pressure: 30 psi) were delivered via a glass capillary positioned 2 cm from the eye to evoke a reliable eye blink reflex in the ipsilateral eye, which was monitored using a video camera (*Figure 2b–d*). This form of sensory stimulation was chosen to evoke dendritic calcium signals in PNs of lobule V, as described previously (*Najafi et al., 2014a*; *Najafi et al., 2014b*). Mice were naïve to the sensory stimulus receiving test stimuli (<5) before recording began and 5–30 stimuli (average of 13 recordings per PN) during a single recording session. During sessions, all air puffs were unexpected, triggering an eye blink reflex in the ipsilateral eye and no conditioned pre-stimulus eye blink responses were observed, indicating that the mouse did not learn to expect the stimulus (*Figure 2—figure supplement 1*).

Sensory-evoked voltage signaling has not been explored in the spiny dendrites of awake animals. Thus, we begin by describing the various dendritic voltage signals evoked by sensory stimulation. Simultaneous voltage and calcium dendritic recordings revealed DCSs, DSs and sustained depolarizing or hyperpolarizing signals during sensory stimulation (*Figure 2e–h* and *Figure 2—figure supplement 3*), similar to those observed following electrical stimulation of PFs. The dendritic signals we observed were not evoked during voluntary eye blinks or detected while the mouse was anesthetized (1% isoflurane) and presented with the same air puff stimuli, confirming their relationship to the sensory stimulation (*Figure 2—figure supplements 2* and *5*, respectively).

DCSs and DSs were often, but not always, detected during coincident depolarizing or hyperpolarizing signals. For example, two PNs, PN1 and PN2, labeled in the same mouse (shown in *Figure 2a*) had different responses to sensory stimulation; PN1 shows a depolarization (black overlay trace) with a DCS riding on top (red trace) (*Figure 2e*), while PN2 shows only a DCS, with no detectable underlying depolarizing or hyperpolarizing signal (*Figure 2f*). The amplitude and direction (i.e., depolarizing or hyperpolarizing) of evoked voltage signals were variable between individual PNs and on a trial-by-trial basis (see *Figure 2e–h* and figure supplements for more examples).

We measured the maximum change in dendritic voltage (excluding DCS and DS events) during the stimulus, as indicated by black arrows in *Figure 2e–h* insets, for all recording trials (n = 696, 54

PNs). We did this by removing all DCSs and DSs and by applying temporal filtering (10 ms boxcar) (see Materials and methods for spikelet detection and removal). A histogram for maximum change in dendritic voltage (excluding spikes) for all trials is shown in *Figure 2i*. Note that combining all trials within our sampling region (*Figure 1a*), the maximum change in dendritic voltage was positive (i. e., depolarizing) (2.91 ± 4.3% ΔF/F, mean ± SD, 54 PNs, t-test, p=7.25 × $10^{-6}$) which corresponds to an average deflection in membrane potential of approximately 6.1 ± 9.0 mV.

By applying a detection threshold of >2 standard deviations above and <2 standard deviations below the baseline voltage signal (baseline calculated in a 500 ms window before the stimulus), we sorted voltage recordings into four groups based on the amplitude and direction of the sensory-evoked voltage response; 'depolarizing', 'no response', 'hyperpolarizing', and 'depolarizing + hyperpolarizing' (*Figure 2j* and *Figure 2—figure supplement 4*). Approximately 53% of voltage recordings displayed a depolarizing signal during the stimulus (max Δ in dendritic voltage > + threshold); 14% showed hyperpolarizing voltage signals (max Δ in dendritic voltage < − threshold); 7% had a truncated signal that appeared to be a superposition of both depolarizing and hyperpolarizing signals (max Δ in dendritic voltage > + threshold, and < − threshold, at different time points during the stimulus); and 26% showed no response during the stimulus (did not reach threshold in either depolarizing or hyperpolarizing directions). The temporal profile of each voltage response was resolved by sorting and averaging voltage signals in each response group (*Figure 2j*). Depolarizing signals increased gradually throughout the stimulus, beginning 9 ± 3 ms (mean ± SD, n = 307) after stimulus onset, and reaching a maximum after 101 ± 3 ms, which is essentially at the stimulus offset. Hyperpolarizing signals were sharper, and their onset occurred later, beginning 35 ± 2 ms (mean ± SD, n = 83) after stimulus onset, and reaching a maximum hyperpolarization 57 ± 2 ms (mean ± SD) after stimulus onset (*Figure 2—figure supplement 4*).

The maximum change in dendritic voltage was similar between depolarizing (6.7 ± 4.7% ΔF/F, mean ± SD, n = 307), hyperpolarizing (6.1 ± 4.0 %ΔF/F, mean ± SD, n = 83), and depolarizing + hyperpolarizing groups (7.0 ± 4.2% ΔF/F, mean ± SD, n = 38) and significantly larger than the no response group (2.6 ± 1.0% ΔF/F, mean ± SD, n = 133), ANOVA, p<1.2 × $10^{-8}$, n = 561, 49 PNs (*Figure 2—figure supplement 4*). The time of eyelid closure (measured from stimulus onset to maximum eyelid closure) was independent of response type (*Figure 2—figure supplement 4*); depolarized (128 ± 40 ms, mean ± SD), hyperpolarized (123 ± 44 ms, mean ± SD), depolarized + hyperpolarized (124 ± 32 ms, mean ± SD), and no response (139 ± 50 ms, mean ± SD) groups, Kruskal–Wallis ANOVA, p>0.35, n = 561, 49 PNs. This confirmed that a lack of dendritic voltage response was not due to a failure of the sensory stimulus, but rather that there was no detectable change in synaptic input to the PN dendrites.

Individual PNs displayed a trend toward a particular voltage response (*Figure 2—figure supplement 4* and see also *Figure 2—figure supplement 6*). The majority of PNs exhibited 'depolarizing' responses (36/54 PNs), and the remaining PNs exhibited either 'hyperpolarizing' (6/54 PNs), both 'depolarizing + hyperpolarizing' (1/54 PNs), or 'no response' (11/54 PNs). The average fraction of recordings across all PNs showing their preferred response was 0.65 ± 0.18 (mean ± SD, n = 54 PNs) (*Figure 2—figure supplement 4*).

It was shown previously that evoked calcium signals in PN dendrites are enhanced compared to spontaneous calcium signals (*Najafi et al., 2014a*; *Najafi et al., 2014b*). Interestingly, we found this only to be true for PNs within the 'depolarizing' voltage response group (depolarizing spontaneous: 13 ± 10% ΔF/F; depolarizing evoked: 19 ± 14%ΔF/F, mean ± SD, ANOVA, p<6 × $10^{-8}$, 36 PNs) and the 'hyperpolarizing' voltage response group (hyperpolarizing spontaneous: 9 ± 7%ΔF/F; hyperpolarizing evoked: 18 ± 12%ΔF/F, mean ± SD, ANOVA, p<$10^{-7}$, 6 PNs), but evoked calcium signals from PNs within the 'no response' group were not enhanced compared to spontaneous calcium signals (no response spontaneous: 11 ± 8%ΔF/F; no response evoked: 13 ± 9% ΔF/F, mean ± SD, ANOVA, p=0.99, 11 PNs) (*Figure 2—figure supplement 4*).

We used pharmacology to confirm that the sensory stimulus-evoked dendritic signals were due to synaptic input. The voltage signals evoked by sensory stimulation were blocked following application of CNQX (100 μM) (*Figure 2k* and *Figure 2—figure supplement 7*) and also Na+ channel blocker Lidocaine (0.2%) (*Figure 2—figure supplement 8*), confirming that these dendritic signals were driven by synaptic input (and not movement artifacts). The maximum change in dendritic voltage (measured within a 200 ms window after stimulus onset) was significantly reduced under CNQX conditions; control (8.0 ± 7.3% ΔF/F, mean ± SD, n = 45), CNQX (4.4 ± 3.4% ΔF/F, mean ± SD, n = 46),

unpaired t-test, p=0.003, n = 91, 5 PNs. The eyelid closure time following CNQX application was not significantly different to control conditions (*Figure 2—figure supplement 7*); control (134 ± 40 ms, mean ± SD, n = 45); CNQX (156 ± 44 ms, mean ± SD, n = 37), Kolmogorov–Smirnov, p=0.12, n = 82, 5 PNs. Application of CNQX and Lidocaine also blocked DCSs and DSs (*Figure 2—figure supplements 7* and *8*). The relationships between the time of maximum change in dendritic voltage (depolarizing or hyperpolarizing) after stimulus onset and the time for eyelid closure under control and CNQX or Lidocaine conditions are shown in *Figure 2—figure supplements 7* and *8*, respectively.

From the observations made thus far we conclude that sensory stimulation evokes graded depolarizing and hyperpolarizing voltage signals in PN spiny dendrites. On average these dendritic voltage signals have a relatively small amplitude (6.1 ± 9.0 mV), below threshold for triggering VGCCs, have a much slower time course compared to DCSs and DSs and are blocked by CNQX (*Figure 2k* and *Figure 2—figure supplement 7*). Since CF synaptic input reliably evokes rapid DCSs (*Roome and Kuhn, 2018*) and noting their similarity to the voltage signals evoked by PF stimulation shown in *Figure 1i–k*, we conclude that these signals originate primarily from PF synaptic input. On average the sensory stimulus-evoked PF input gives rise to a depolarizing signal across the PN dendrites (global depolarization) that begins 9 ± 3 ms after stimulus onset, which is consistent with rapid and dense GC responses to sensory stimulation (*Giovannucci et al., 2017*; *Jörntell and Ekerot, 2006*). The sensory stimulation-evoked hyperpolarizing dendritic signals are also in agreement with an increase in MLI activity, and feed forward inhibition evoked by sensory stimulation (*Chu et al., 2012*; *Gaffield et al., 2018*). The trial-by-trial variability in PN dendritic voltage responses that we detect is likely to result from an interplay between these two synaptic inputs driven by the MF pathway.

## Sensory stimulation evokes graded voltage-dependent dendritic calcium signals in absence of DCSs and DSs

In several recording trials we detected sustained voltage signals (depolarizing and hyperpolarizing) during sensory stimulation with no coincident DCSs or DSs (see *Figure 3a* for example). While dendritic calcium signals in PNs are predominantly mediated by P/Q-type VGCCs triggered during DCSs or DSs (*Usowicz et al., 1992*), additional mechanisms also contribute to dendritic calcium signaling at PF–PN synapses, which do not require DCS or DS events. For example, glutamate released during PF input can activate mGluR1s and trigger calcium influx (*Tempia et al., 2001*) and modulate low threshold T-type VGCCs (*Ait Ouares and Canepari, 2020*; *Hildebrand et al., 2009*; *Otsu et al., 2014*), and also trigger calcium release from intracellular stores (*Wang et al., 2000*). Whether or not these signals are involved in dendritic signaling in PNs during sensory stimulation has not been confirmed.

To determine if dendritic calcium signals can be evoked by sensory stimulation in the absence of CF input, we selected all recordings in which no DCSs or DSs were detected in a window ±300 ms of the stimulus onset. Averaging these recordings revealed a small but clear increase in dendritic calcium during the sensory stimulation that reached a maximum 135.8 ± 48.3 ms after stimulus onset (dendritic calcium at stimulus offset relative to baseline: 1.8 ± 2.8% ΔF/F, mean ± SD, n = 59, t-test, p=5.9 × 10$^{-6}$) (*Figure 3b*), which was blocked following application of CNQX (100 μM) (*Figure 3c*); average calcium at stimulation offset under CNQX conditions was −0.2 ± 2.3% ΔF/F, mean ± SD, and not significantly different from baseline, t-test, p=0.59, n = 36, 5 PNs.

Calculating the average voltage and calcium signal over the full sensory stimulus, on a trial-by-trial basis, revealed the voltage–calcium relationship (n = 59, Pearson's r = 0.71, p=4.51 × 10$^{-10}$) (*Figure 3d*). This confirms that PF-evoked dendritic voltage signals (described in *Figure 2*) generate voltage-dependent dendritic calcium signals in the absence of CF input.

## DCSs and DSs triggered by coincident CF and PF synaptic input evoke supralinear calcium signals during sensory stimulation

Sensory stimulation is known to trigger enhanced calcium signals in PN dendrites of the cerebellar vermis, compared to spontaneous calcium signals (*Najafi et al., 2014a*; *Najafi et al., 2014b*). While various mechanisms have been shown to modulate dendritic calcium signals in PNs, including MLI activity (*Callaway et al., 1995*; *Gaffield et al., 2018*; *Kitamura and Häusser, 2011*), PF activity

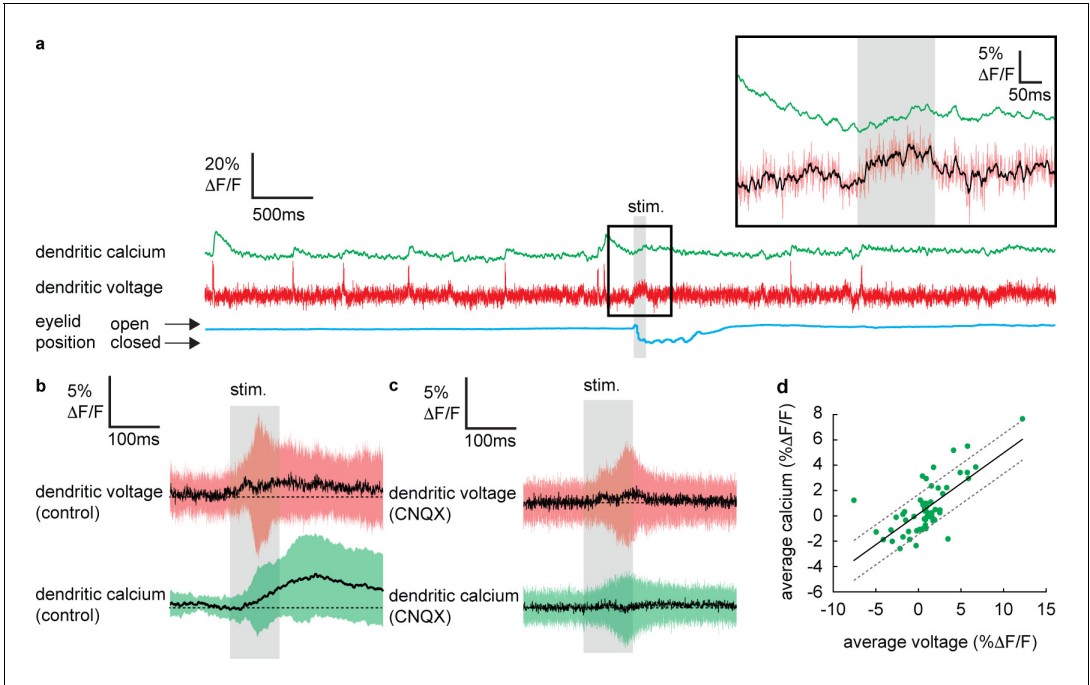

**Figure 3.** Sensory stimulation evokes graded voltage-dependent dendritic calcium signals in the absence of dendritic complex spikes (DCSs) and dendritic spikes (DS). (a) Single trial recordings of simultaneous dendritic voltage (red traces) and calcium (green traces). DCS and DS events are visible in both voltage and calcium traces. Blue traces show average intensity of region of interest (ROI) used to record eye responses. Gray bars show full duration of air puff stimulus. Inset shows zoomed voltage traces during the sensory stimulus. The voltage recording in inset shows a depolarizing voltage signal but no DCSs or DSs during the air puff (black trace shows 10 ms boxcar filtered voltage recording). (b and c) Averages of dendritic voltage (red) and calcium (green) during sensory stimulation (only recordings with no dendritic spikelets and both depolarizing and hyperpolarizing signals are included), under control (b) and CNQX (c) conditions; black lines show mean, colored traces show SD. (d) Relationship between average dendritic voltage and calcium during the stimulus, solid line is linear regression (Pearson's r = 0.71, p=4.51 $\times$ 10$^{-10}$) and dashed lines show 95% confidence intervals.

(*Wang et al., 2000*), and graded CF input (*Gaffield et al., 2019*; *Roh et al., 2020*), the voltage signals that trigger these calcium signals have not been directly explored in awake animals. Thus, we next examined the DCSs and DSs evoked by sensory stimulation (*Figure 4*).

As described previously (*Roome and Kuhn, 2018*), we detected two types of spike events in the PN spiny dendrites; CF-evoked DCS, which occurred at a reliable average rate of 1.05 ± 0.4 Hz (mean ± SD, 54 PNs, 696 recordings) and PF-evoked DS, which occurred irregularly at an average rate of 0.13 ± 0.3 Hz (mean ± SD, 54 PNs, 696 recordings). Importantly, we detected an increase in frequency of both DCSs and DSs during the sensory stimulus (*Figure 4a*). Peri-stimulus time histograms (PSTH) for DCS or DS events (*Figure 4b*) show how their frequency increased dramatically during the sensory stimulus. Average DCS and DS frequencies during the stimulus was 5.73 ± 5.51 Hz and 1.24 ± 0.77 Hz (mean ± SD), respectively; a relative increase in DCS and DS frequency of 5.5 and 9.5 times, respectively (see *Figure 4b* inset). The average timing of DCS and DS signals after stimulus onset was 43.3 ± 22.1 ms (mean ± SD, n = 414) and 53.3 ± 24.5 ms (mean ± SD, n = 92) respectively.

We compared DCS and DS events in a pre-stimulus time window (100 ms window prior to stimulus onset, see *Figure 5b* inset) with the stimulus time window, in all recordings and calculated the probability for the following outcomes: 'no DCS, DS', 'DCS only', 'DS only', and paired 'DCS + DS' (*Figure 4c*). There was a distinct increase in the probability of detecting DCSs and DSs during the stimulus compared to the pre-stimulus time window; DCS only (pre-stimulus: 11.5 ± 1.2%, stimulus: 49.3 ± 1.7%); DS only (pre-stimulus: 0.8 ± 0.4%, stimulus: 7.6 ± 1.1%); paired DCS + DS signals (pre-stimulus: 0.3 ± 0.2%, stimulus: 5.9 ± 0.9%) (mean ± SD, n = 688, Kolmogorov–Smirnov, p=7.3 $\times$ 10$^{-79}$).

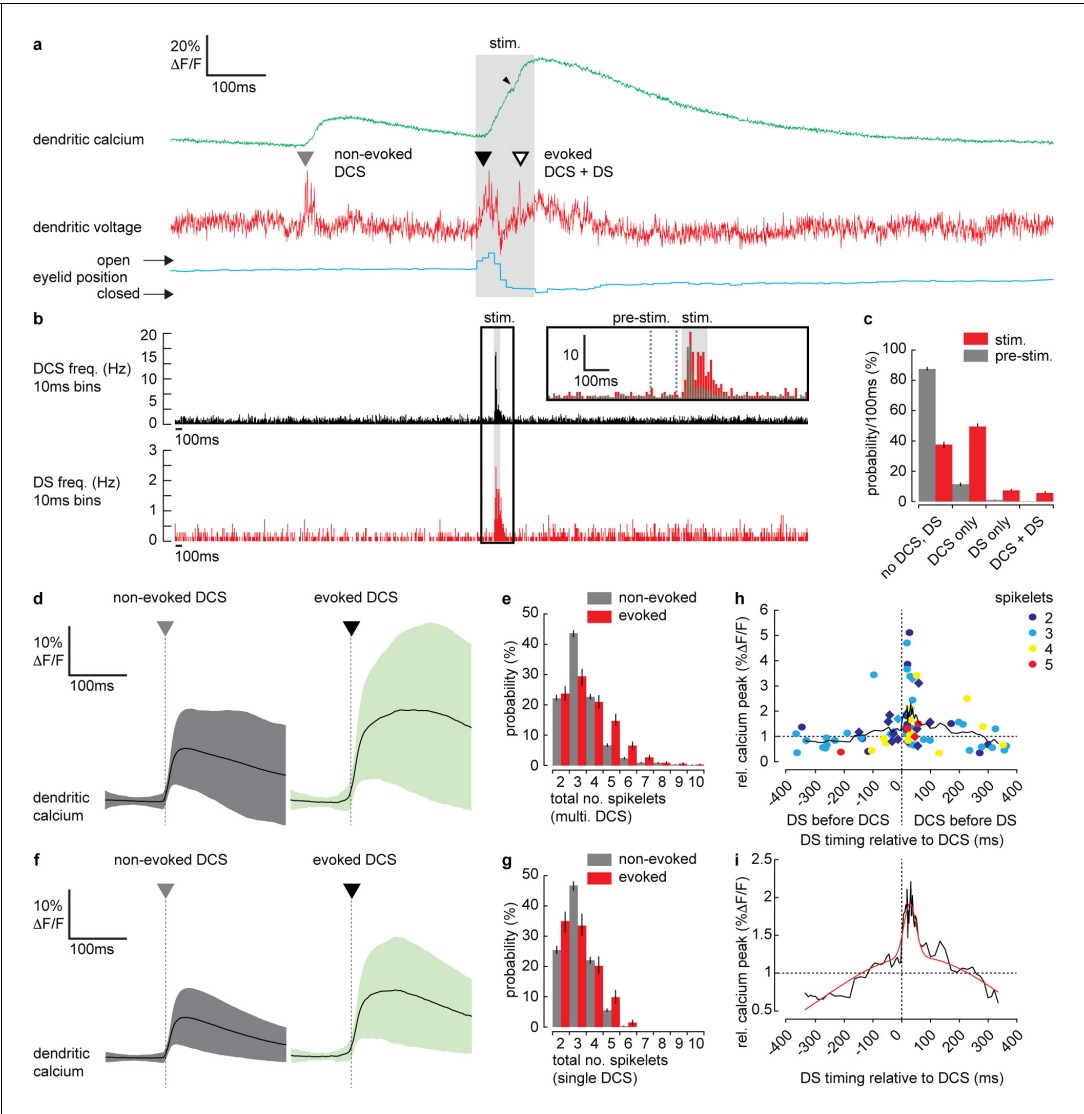

**Figure 4.** Dendritic complex spikes (DCSs) and dendritic spikes (DSs) triggered by coincident climbing fiber (CF) and parallel fiber (PF) synaptic input evoke supralinear calcium signals during sensory stimulation. (**a**) A simultaneous recording of dendritic calcium (green trace) and voltage (red trace) during air puff sensory stimulation, showing dendritic complex spike (DCS; filled triangle) and dendritic spike (DS; open triangle) signals. Signals that occur during the air puff stimulus (stim.) are termed 'evoked' (black triangle), otherwise 'non-evoked' (gray triangle). Small black arrowhead indicates increase in calcium signal following the DS. Corresponding eyelid response (blue trace) and timing of air puff stimulus (gray bar). (**b**) PSTH for DCS frequency (top black) and DS frequency (bottom red) using 10 ms bins. Inset shows zoomed overlay of DCS (black) and DS (red) frequencies, relative to their average frequencies. Dashed gray lines indicate the pre-stimulus period (100 ms window prior to air puff onset). (**c**) Probability distributions for all possible outcomes; 'no DCS, DS', 'DCS only', 'DS only', and paired 'DCS + DS', during pre-stimulus and stimulus time windows. (**d**) Averages of all calcium signals following non-evoked (gray, left) and evoked (green, right) DCS signals. Black trace is mean and colored trace is SD. (**e**) Probability distributions for the total number of dendritic spikelets detected in a 100 ms window following onset of each non-evoked and evoked DCS signal, including multiple DCS signals. (**f**) Averages of all calcium signals following non-evoked (gray, left) and evoked (green, right) DCS signals, excluding multiple DCS signals. Black trace is mean and colored trace is SD. (**g**) Probability distributions for the total number of dendritic spikelets detected in a 100 ms window following onset of each non-evoked and evoked DCS signal, excluding multiple DCS signals. (**h**) Relationship between DCS + DS pair timing and the corresponding relative calcium peak. Relative calcium peaks were calculated with respect to the average calcium signal of DCS with the same number of spikelets (but with no paired DS signal). Colors represent the number of DCS spikelets for each DCS + DS pair. Data points for evoked DCS + DS pairs are diamonds and non-evoked DCS + DS pairs are circles. Black line shows boxcar average of 10 consecutive data points and (**i**) two-term Gaussian fit (red line) applied to boxcar average (adjusted r-square = 0.9117). Vertical bars in probability distributions show bootstrapped SD. The online version of this article includes the following figure supplement(s) for figure 4:

**Figure supplement 1.** Types of dendritic voltage and calcium signals in Purkinje neurons (PNs) of awake mice, resolved with fast (2 kHz) simultaneous voltage and calcium imaging.

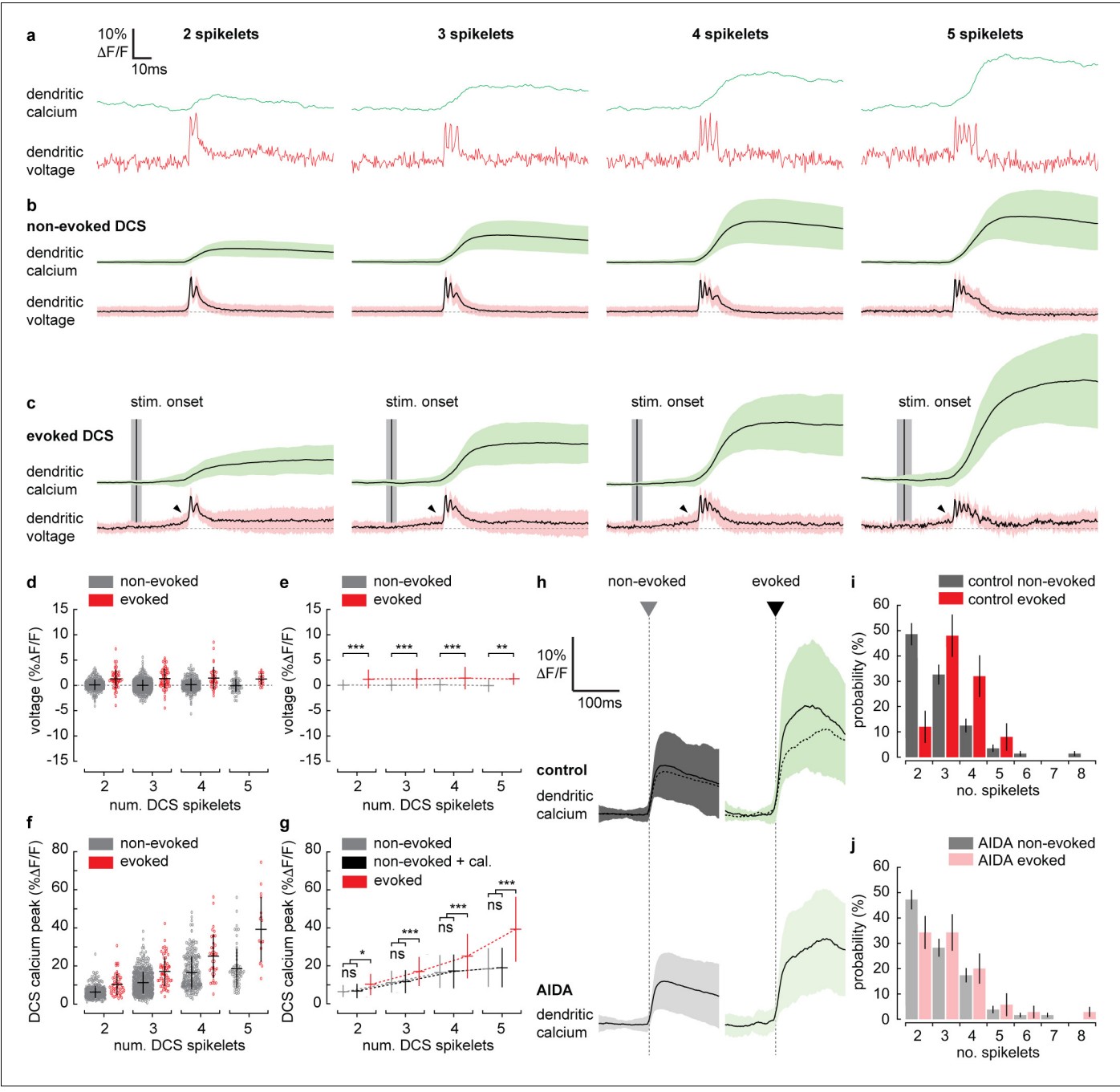

**Figure 5.** Supralinear dendritic signaling in Purkinje neuron (PN) dendrites during sensory-evoked stimulation is voltage and mGluR1 dependent. (**a**) Single trial examples of non-evoked dendritic complex spike (DCS) signals sorted based on the number of spikelets showing dendritic voltage (red traces) and calcium (green traces). (**b**) Averages of all non-evoked DCS signals. (**c**) Averages of all evoked DCS signals. Arrowheads indicate depolarization preceding evoked DCS signals. Mean time of sensory stimulation onset (stim. onset) is shown by vertical black line and vertical gray bar shows SEM. Full stimulus duration is 100 ms. (**d**) Average pre-DCS voltage for evoked (red) and non-evoked (gray) signals, grouped based on the number of DCS spikelets. Horizontal bars show mean, vertical bars show SD. (**e**) Mean from all groups in (**d**). (**f**) Average calcium peak for evoked (red) and non-evoked (gray) signals, grouped based on the number of DCS spikelets. (**g**) Mean from all groups in (**f**) and sum of non-evoked and + cal. groups (black). (**h**) Averages of all calcium signals following non-evoked (gray, left) and evoked (green, right) DCS signals, under control (top dark traces) and mGluR1 antagonist (AIDA) conditions (bottom light traces and top overlaid dashed traces). Black trace is mean and colored trace is SD. Probability distributions for the total number of dendritic spikelets detected in a 100 ms window following onset of each non-evoked and evoked DCS signal, excluding multiple DCS signals, under (**i**) control and (**j**) AIDA conditions (Kolmogorov–Smirnov, vertical bars in probability distributions show bootstrapped SD). *p<0.05, **p<0.01, and ***p<0.001.

*Figure 5 continued on next page*

*Figure 5 continued*

The online version of this article includes the following figure supplement(s) for figure 5:

**Figure supplement 1.** mGluR1 antagonist, AIDA, has no effect on evoked eye blink closure time, stimulus-evoked dendritic voltage signaling, or the probability of dendritic complex spike (DCS) and dendritic spike (DS) events.

In agreement with previous studies (*Gaffield et al., 2019*; *Najafi et al., 2014a*; *Najafi et al., 2014b*), we found that on average, evoked calcium signals (i.e., signals triggered during the stimulus) were enhanced relative to non-evoked calcium signals (i.e. signals occurring before the stimulus) (relative calcium peak; $1.4 \pm 0.6$, 51 PNs, 365 recordings, t-test, p=$1.5 \times 10^{-5}$) (*Figure 4—figure supplement 1*). To determine the mechanisms underlying the enhanced calcium signals we examined the individual DCS and DS events. DCSs, DSs and corresponding calcium signals detected during the stimulus (termed 'evoked' signals), were compared with equivalent signals detected prior to the stimulus (termed 'non-evoked' signals). We counted the total number of spikelets detected in a 100 ms window following each evoked or non-evoked DCS signal, including any additional DCS and DS signals within that time window, and measured the resulting calcium signal peak. Calcium peaks were on average larger following evoked DCS signals compared to the non-evoked DCS signals (non-evoked: $12.6 \pm 9.3\%$ ΔF/F, n = 1624, evoked: $20.8 \pm 15.1\%$ ΔF/F, n = 333, mean ± SD, unpaired t-test, p=$1.6 \times 10^{-37}$) (*Figure 4d*). Accordingly, the corresponding dendritic voltage recordings showed an increase in the total number of dendritic spikelets following each evoked DCS signal compared to non-evoked DCS signals, evident through a positive shift in the spikelet count probability distribution (Kolmogorov–Smirnov, p=$1.85 \times 10^{-5}$) (*Figure 4e*).

We considered the possibility that the increase in dendritic calcium peak and the number of spikelets result from an increase in the number of CF inputs (i.e., multiple DCS signals) occurring within the 100 ms time window. Due to slow calcium indicator dynamics, high frequency CF inputs cannot always be resolved from the calcium signal alone (see *Figure 4—figure supplement 1*). Using the voltage recording we detected and excluded all signals in which multiple CF inputs were observed and repeated the analysis. Interestingly, when only single DCS signals were permitted, the peak calcium of evoked signals remained enhanced compared to non-evoked signals (non-evoked: $11.9 \pm 8.1\%$ΔF/F, n = 1292, evoked: $19.7 \pm 14.2\%$ΔF/F, n = 203, mean ± SD, unpaired t-test, p=$3.8 \times 10^{-28}$) (*Figure 4f*). The time to peak for evoked calcium signals was also delayed compared to non-evoked calcium signals (non-evoked: $37.7 \pm 26.3$ ms, n = 1292, evoked: $69.0 \pm 44.9$ ms, n = 203, mean ± SD, unpaired t-test, p=$2.7 \times 10^{-42}$) (see *Figure 4—figure supplement 1*, for a comparison of all dendritic calcium signals described so far). However, the difference between pre-stimulus and stimulus probability distributions of spikelet numbers was no longer significant (Kolmogorov–Smirnov, p=0.075) (*Figure 4g*). These differences in the calcium signal time-course are indicative of an additional source of calcium following the evoked DCS, causing a delayed and enhanced calcium peak, while the similarity in spikelet numbers between evoked and non-evoked DCS suggests this additional calcium source is spike-independent. Thus, the enhanced calcium signal is not due to an increase in CF input frequency, or an increase in dendritic spiking, caused by stronger CF input, but through an additional mechanism that does not require DCSs or DSs, such as described in *Figure 3*, or potentially calcium released from internal stores (*Llano et al., 1994*; *Llano et al., 1991a*; *Wang et al., 2000*).

Since both DCS and DS frequency increased during the sensory stimulus, we next considered the possibility that enhanced evoked calcium signals result from paired DCS and DS signals that occur during the same stimulus. We investigated the relationship between the timing of paired DCS + DS signals and their corresponding dendritic calcium signal, by selecting all DCS + DS pairs (non-evoked and evoked) with <400 ms temporal separation. We measured the calcium peak following each pair of DCS + DS signals, relative to the average calcium peak of a single DCS signal comprising the same number of spikelets. Relative calcium peaks were plotted in relation to the temporal separation of each DCS + DS pair (*Figure 4h*). Analysis revealed a clear asymmetric and supralinear relationship between the timing of DCS and DS signals and their corresponding calcium signals. The relative calcium peak was maximum when DS followed DCS events and was independent of the number of DCS spikelets (two spikelets: $1.8 \pm 1.3$ [n = 13]; three spikelets: $2.0 \pm 1.2$ [n = 15]; four spikelets: $1.7 \pm 0.9$ [n = 6]; five spikelets: $1.3 \pm 0.3$ [n = 3], mean ± SD, measured in the 80 ms time window

following the DCS, ANOVA, p>0.7763). By fitting a two-term Gaussian function (*Figure 4i*) we estimate that the average relative calcium peak (2.0 ± 0.1) is maximum when DS follow DCS signals by 28.3 ± 5.4 ms (mean ± SD). This means that an additional DS (evoked by PF input) following a DCS (evoked by CF input) can enhance the dendritic calcium signal supralinearly (potentially by more than 100% relative to the DCS event alone) depending on the timing of the two events.

Thus, we conclude that paired DCS + DS signals evoked by coincident CF and PF input act to enhance the dendritic calcium signal supralinearly, and this enhancement is dependent on the timing of paired DCS + DS events. However, paired DCS + DS signals are relatively rare (detected in only 5.9 ± 0.9% of recordings), suggesting an additional coincidence detection mechanism at work. Taken together, our findings thus far reveal a predominantly depolarizing voltage signal and an increase in DCS and DS activity during sensory stimulation. Both these signals are blocked by CNQX and so originate primarily from coincident excitatory PF and CF synaptic input.

## Supralinear dendritic signaling in PN dendrites during sensory-evoked stimulation is voltage and mGluR1 dependent

To investigate the mechanisms underlying the evoked supralinear calcium signals further, we selected individual evoked and non-evoked DCS signals only (now with no subsequent DCS or DS signals). We sorted DCS signals based on the number of spikelets generated. This created four DCS groups, ranging from 2 to 5 spikelets (*Figure 5a–c*). Averaging voltage traces within each evoked spikelet group revealed a prominent depolarization beginning before the evoked DCS event; this 'pre-DCS' signal was not detected before non-evoked DCS signals however (see black arrowheads in *Figure 5c*). This would be expected because, on average, sensory stimulation-evoked voltage signals are depolarizing, and their onset precedes DCS generation (as determined in *Figures 2* and *4*). Average pre-DCS voltages (measured in a 10 ms window preceding DCS signals) for each group are shown in *Figure 5d* (evoked pre-DCS vs. non-evoked pre-DCS, ANOVA, p<0.004). The evoked pre-DCS depolarization amplitude was independent to the number of DCS spikelets (two spikelets: 1.31 ± 1.84% ΔF/F; three spikelets: 1.35 ± 1.93% ΔF/F; four spikelets: 1.47 ± 2.22% ΔF/F; five spikelets: 1.29 ± 1.14% ΔF/F, mean ± SD, ANOVA, p>0.52) (*Figure 5e*).

We would expect the pre-DCS depolarization detected during evoked signals to contribute to the calcium signal through the voltage-dependent increase in calcium baseline that we described in *Figure 3*. Using the voltage-calcium relationship determined in *Figure 3*, and the average pre-DCS voltage across all spikelet groups; 1.36 ± 1.89% ΔF/F (approximately 2.9 ± 4.0 mV, mean ± SD), we would predict an average increase in calcium baseline of 0.8 ± 1.5% ΔF/F.

The average calcium peak, however, is much larger during evoked DCS signals compared to non-evoked DCS signals comprising the same number of spikelets. Furthermore, the difference between evoked and non-evoked calcium peaks increased supralinearly with the number of DCS spikelets (non-evoked vs. evoked, two spikelets: 6.2 ± 3.2% ΔF/F vs. 10.1 ± 5.5% ΔF/F; three spikelets: 11.1 ± 5.7% ΔF/F vs. 17.0 ± 7.7% ΔF/F; four spikelets: 16.4 ± 8.6% ΔF/F vs. 25.1 ± 11.7% ΔF/F; five spikelets: 18.5 ± 10.2% ΔF/F vs. 39.3 ± 17.1% ΔF/F, mean ± SD, ANOVA, p<0.002) (*Figure 5f*). In this case, the linear sum of non-evoked DCS calcium peak and our predicted increase in calcium baseline ('+ cal.') could not account for the supralinear enhancement in calcium peak during evoked DCS signals (non-evoked + cal. vs. evoked, ANOVA, p<0.043) (*Figure 5g*).

Activation of mGluR1s by coincident PF and CF input is known to drive supralinear calcium signals at PF–PN synapses in cerebellar slices (*Wang et al., 2000*), but their contribution to PN dendritic signaling has not been explored in vivo. By application of AIDA (a potent and selective mGluR1 antagonist), we assessed the contribution of mGluR1 to sensory-evoked dendritic signals in the awake mouse. After application of AIDA, sensory-evoked calcium signals were significantly reduced (control: 24.4 ± 12.3% ΔF/F, AIDA: 17.9 ± 9.3% ΔF/F, mean ± SD, unpaired t-test, p=0.025) (*Figure 5h*). Non-evoked calcium signals, however, were not significantly altered (control: 10.6 ± 6.5% ΔF/F, AIDA: 9.5 ± 5.5% ΔF/F, mean ± SD, unpaired t-test, p=0.116), indicating that the contribution of mGluR1 activity to the calcium signal is specific to sensory-evoked calcium signals when CF and PF inputs coincide, and we propose that mGluR1s are activated at PF–PN synapses, where the PF and CF synaptic potentials converge.

Probability distributions for DCSs and DSs detected in pre-stimulus and stimulus time windows were calculated under control and AIDA conditions. As in *Figure 4c*, there was a clear increase in signal probability during the stimulus compared to pre-stimulus time windows under both control

and AIDA conditions, but we detected no difference between control vs. AIDA conditions (for either pre-stimulus or stimulus groups), indicating that the mGluR1 antagonist did not directly affect CF and PF input (*Figure 5—figure supplement 1*). There was also no significant effect of AIDA on evoked dendritic voltage signaling, supporting the conclusion that these signals are predominantly AMPA mediated (*Figure 5—figure supplement 1*).

However, we found that the probability distribution for the total number of spikelets following evoked DCS was shifted toward lower numbers under AIDA conditions. Consequently, unlike control conditions (*Figure 5i*), the probability distributions for the total number of spikelets following evoked and non-evoked DCS signals were no longer significantly different following AIDA application (*Figure 5j*). This suggests that mGluR1 activity enhances spikelet generation during evoked DCS signals.

## Discussion

Dendritic coincidence detection is a salient mechanism in neuronal processing. We used simultaneous voltage and calcium two-photon imaging of single PN dendrites at high temporal resolution (2 kHz) to explore PF and CF coincident input during sensory-evoked responses. Due to technical limitations, PF-evoked input to PN dendrites – that are predominantly voltage signals – have been thus far undetectable in vivo. As such, confirming physiological interaction between PF and CF input has not been possible, and previous in vivo studies have instead focused on coincident MLI and CF activity and dendritic calcium signals (*Callaway et al., 1995*; *Gaffield et al., 2018*; *Kitamura and Häusser, 2011*). The ability to distinguish between depolarizing and hyperpolarizing dendritic voltage signals and DSs or DCSs evoked by either PF or CF input provides fresh insight into how dendritic calcium signals triggered by sensory stimulation are modulated, and how dendritic coincident detection of the PF and CF input pathways occur in vivo.

### Dendritic coincidence detection mechanisms modulate sensory stimulation-evoked dendritic calcium signals in PN dendrites

Here we show that sensory stimulus-evoked instructive signals carried to the cerebellum via CFs and MFs converge onto the dendrites of single PNs, and thereby modulate the dendritic calcium response, in an individual PN selective manner (compare responses from nearby PNs in *Figure 2*), supporting predictions for a high degree of spatial convergence for these input pathways (*Apps and Hawkes, 2009*; *Brown and Bower, 2001*; *Garwicz et al., 1998*; *Odeh et al., 2005*; *Pijpers et al., 2006*).

PF and CF synaptic input is integrated at the PN spiny dendrites during sensory-evoked stimulation in awake mice. To summarize, we identify three interplaying dendritic coincidence detection mechanisms that modulate calcium signals at the PN dendritic level.

PN dendrites integrate EPSPs from PF input (*Figure 2*), which can result in a graded (voltage-dependent) calcium signal (*Figure 3*; cf. *Figure 1*). In some trials we also detected a sensory stimulus-evoked hyperpolarizing dendritic voltage signal. We propose these are due to feed forward inhibition via the PF-MLI-PN pathway (*Mittmann et al., 2005*). We expect that the voltage-dependent dendritic calcium signal is continually regulated by EPSP and IPSP and the first mechanism of dendritic coincidence detection occurs when this voltage-dependent calcium signal coincides with a CF-evoked calcium signal.

The second mechanism is through the PF-evoked DS events, whose frequency increases during the sensory stimulus, and when paired with DCS act to enhance the CF-evoked calcium signal in a time-dependent manner (*Figure 4*). We found the greatest enhancement in calcium signal occurred when the DS followed the DCS by ~28 ms, which appears to be in contrast to in vitro experiments, where the maximal enhancement in calcium signal was detected when PF stimulation preceded CF stimulation (*Wang et al., 2000*). However, we note that the DS events are due to a culmination of integrated PF input, and so the timing of the DS event does not indicate the onset of PF input, but the moment that the threshold in membrane potential is reached to trigger the DS event, and so it is expected that PF input onset begins sooner. Indeed, we found that on average PF input onset precedes CF input during sensory stimulation by ~34 ms, as shown in *Figures 2j* and *Figure 4b*, and as indicated by the pre-DCS depolarization in *Figure 5c*. While PF-evoked DS signals are relatively rare in awake mice (*Roome and Kuhn, 2018*), we found that their frequency increased 9.5 times

during sensory stimulation, and their impact on the dendritic calcium signal was supralinear. Thus, we propose PF-evoked DS signals play a unique role in modulating dendritic calcium signals during sensory stimulation. In fact, the additional PF-evoked DS could be viewed as a correction to the CF-evoked DCS signal, supplementing an otherwise 'too weak' DCS signal with extra DSs.

The third mechanism we describe is through mGluR1 activity (*Figure 5*), which is known to trigger supralinear calcium signals when PF and CF inputs are conjunctively activated, and thereby modify PF–PN synaptic strength (*Sarkisov and Wang, 2008*; *Wang et al., 2000*). It was interesting to find that mGluR1 activity can directly influence DCS generation by the CF input and the time-course of the resulting dendritic calcium signal, specifically during evoked DCS events (c.f. *Figure 5h and j*). Such an interaction between mGluR1 activity and CF-evoked spikelet generation has previously been postulated (*Batchelor and Garthwaite, 1997*; *Tempia et al., 2001*) and is likely mediated through their interaction with T and P/Q-type VGCCs (*Hildebrand et al., 2009*; *Otsu et al., 2014*).

## Functional implications: A solution to the 'signal-to-noise' problem

The CF input has historically been viewed as an 'all-or-none' type event (*Eccles et al., 1966*), such that CF-evoked EPSPs are identical from one CF input to the next (provided the PN membrane potential is held constant). This description raised a critical problem, however, regarding the proposed role of CFs in providing PNs with instructive 'teaching' signals for motor learning. This problem, referred to as the 'signal-to-noise' problem (*Llinás et al., 1997*), asserts that if PN dendritic responses to CF were identical, 'evoked' and 'spontaneous' CF inputs would be indistinguishable at the level of the PN dendrite. Given that CF inputs are delivered continuously at a rate of ~1 Hz, the information encoded by 'teaching' signals should be lost among the noise of spontaneous CF input (see *Najafi and Medina, 2013* and *Zang and De Schutter, 2019* for review).

A growing body of evidence, however, has proved that CFs are not such inefficient teachers, showing that CF-evoked dendritic calcium signals are in fact graded, and that 'evoked' calcium signals are enhanced compared to 'spontaneous' signals. There have been differing views regarding the underlying mechanisms, however, which could not be directly resolved since it was not possible to measure the PN membrane potential at the location of the calcium signals; the spiny dendrites (*Gaffield et al., 2019*; *Gaffield et al., 2018*; *Najafi et al., 2014a*; *Najafi et al., 2014b*; *Roh et al., 2020*). In contrast to our findings, a recent study has proposed that calcium signals in PN dendrites evoked by sensory stimulation are graded entirely by the strength of CF input, essentially from outside of the cerebellum, without direct contribution from the MF pathway (*Gaffield et al., 2019*). We agree that the strength of CF input has a powerful influence on spikelet generation in PN dendrites (*Davie et al., 2008*) and consequently on dendritic calcium signals (*Gaffield et al., 2019*; *Roh et al., 2020*; *Roome and Kuhn, 2018*), and it cannot be excluded that differences in recording location and/or sensory stimulation may favor differing mechanisms for modulating calcium signals in PN dendrites. However, as seen in *Figure 5*, we find a clear nonlinear relationship between CF-evoked DCSs and the resulting dendritic calcium signal. Importantly, this nonlinear voltage–calcium relationship was specific to CF input evoked by sensory stimulation.

Here we show how coincident PF and CF input evokes supralinear dendritic calcium signals via mechanisms within the PN dendrite; through VGCCs (PF-evoked DS events), and activation of mGluR1s. Independent of the CF input, these coincidence detection mechanisms can modulate the downstream dendritic calcium signal and we propose this significantly enhances the information that can be transmitted by the evoked CF event. In the case where evoked calcium signals in PN dendrites are graded entirely by the strength of CF input, we would expect that their dynamic range would suffer from a similar 'signal-to-noise' problem, such that there remains a limited range in the information that can be transmitted or 'taught' to a single PN. The range of PN dendritic responses would remain limited by the strength of CF input, or more precisely by the number of dendritic calcium spikelets evoked by CF input (typically two to five spikelets). Furthermore, this would suggest that many PF–PN synapses receive the same teaching signal, independent of any instructive signals carried via the PFs, because the CF input propagates actively into the spiny dendrites (*Roome and Kuhn, 2018*). Thus, the range of learning through synaptic plasticity at PF–PN synapses might also face similar limitations.

An important functional implication of the coincidence detection mechanisms described here is that evoked dendritic calcium signals can be adjusted at the PN dendritic level, within the cerebellar circuitry, in a way that accounts for instructive signals carried by MFs via PF input. In principle

dendritic calcium signals can be fine-tuned with higher temporal and spatial precision, potentially at the level of single spines and branchlets throughout the entire dendritic tree (*Denk et al., 1995*), in far more detail than could be attained through enhanced CF input alone, as predicted by computational approaches (*Zang and De Schutter, 2019*; *Zang et al., 2018*). Indeed, spontaneous CF events show a large degree of spatiotemporal variability across dendritic segments in awake mice (*Roome and Kuhn, 2018*) and we predict the spatiotemporal patterns of evoked DCS, modulated by PF input, to be particularly important in conveying information encoded by coincident PF and CF input events, during sensory stimulation. The combination of the various dendritic mechanisms we describe here may function to control the size and spread of the dendritic response to the coincidence detection event (the dendritic calcium signal in this case), thereby defining the 'dendritic computational unit' (*Zang et al., 2018*) and potentially, the extent of synaptic plasticity at PF–PN synapses. This could be achieved through generation of PF-evoked DS events, for example, that actively propagate across multiple dendritic branchlets more effectively than passive PF-evoked EPSPs (*De Schutter and Bower, 1994*; *Roome and Kuhn, 2018*).

It is intriguing that the coincidence detection processes and supralinear dendritic calcium signals we describe here are triggered by sensory stimulation yet occur seemingly continuously; during sensory stimulation in awake naïve mice, and not only during motor learning of a specific task, such as eye blink conditioning. In vitro studies have shown how membrane potential and timing of coincident PF and CF synaptic inputs are critical for modulating calcium signaling and synaptic plasticity at PF–PN synapses (*Ly et al., 2016*; *Piochon et al., 2012*; *Sarkisov and Wang, 2008*). It remains to be seen if supralinear dendritic calcium signals evoked during unconditioned sensory stimulation result in lasting changes in PF–PN synaptic strength, but we might expect to see an equally fluid and variable range of synaptic plasticity at PF–PN synapses during behavior. PN dendrites and their spines are highly sensitive to calcium levels, which can in turn determine the direction of synaptic plasticity at PF–PN synapses (*Coesmans et al., 2004*; *Jörntell and Hansel, 2006*). We expect that to constantly fine-tune plasticity at PF–PN synapses, calcium signals are continuously graded at the global and subcellular dendritic level.

## Conclusion

The coincident activity of CF and PF synaptic input (and PF driven feed forward inhibition via MLIs [*Gaffield et al., 2018*; *Kitamura and Häusser, 2011*]) combined with intrinsic dendritic mechanisms (VGCCs and mGluR1s) allow for a full dynamic spatiotemporal range in dendritic calcium signaling, which we propose serve as efficient mechanisms for decoding the information transmitted by CF-evoked 'teaching' signals. Here, we establish that the combined activity of these mechanisms becomes more apparent during heightened sensory input, as indicated by their enhanced cooperation during sensory-evoked eye blink responses.

## Materials and methods

### Animals and surgery

All animal procedures were conducted in accordance with guidelines of the Okinawa Institute of Science and Technology Institutional Animal Care and Use Committee in an Association for Assessment and Accreditation of Laboratory Animal Care (AAALAC International)-accredited facility, under protocol numbers: 2016–170, 2019–279. Cerebellar chronic cranial window surgeries were performed on 2-month-old male C57/BL6 mice, using a 5 mm glass cover slip with silicone access port (*Kuhn and Roome, 2019*; *Roome and Kuhn, 2014*; *Roome and Kuhn, 2019*). The window was positioned to allow imaging within lobule V of the cerebellar vermis and the access port was positioned to allow access to the imaging area via a micropipette for PN labeling and drug delivery.

### Microscope setup

We used a custom-built combined wide-field, two-photon microscope (MOM, Sutter Instruments) with either a ×5/N.A. 0.13 air objective (Zeiss) or a ×25/N.A. 1.05 water immersion objective with 2 mm working distance (Olympus) with ScanImage software. Bright field imaging was performed using a sCMOS camera (PCO.edge, PCO). A femtosecond-pulsed Ti:sapphire laser (Vision II, Coherent), circularly polarized and under-filling the back focal plane of the ×25 objective, was used to excite

fluorescence (typical laser power at 1020 nm: 60 mW), which was detected by two GaAsP photomultiplier tubes (Hamamatsu) in the spectral range of 490–550 nm (green: GCaMP6f) and 550–750 nm (red: ANNINE-6plus).

The mice were headfixed on a platform that consisted of a vertically rotating treadmill, headplate stage, and micromanipulator tower (Sutter Instruments), all mounted on a horizontally rotating stage (8MR190-90-4247, Standa). The sCMOS camera and infrared light source were used to record behavioral activity during recording at 100 fps. A second infrared video camera (Sony) was used to monitor mouse behavior throughout the experiment. The micromanipulator (M-285, Sutter instruments) was used for GCaMP6f virus (UPenn Vector Core) and ANNINE-6plus dye (http://www.sensitivefarbstoffe.de, Dr. Hinner and Dr. Hübener Sensitive Dyes GbR) injection and also for drug delivery and electrophysiological recording via micropipette.

## Associated viral vector injections

One week following surgery, mice were anesthetized (1–2% isoflurane) and head mounted for two-photon guided injection of the adeno-associated viral vectors (AAVs) into the PN layer approximately 150 μm below the dura. For this, beveled quartz electrodes (0.7 mm ID, pulled and beveled to 10–20 μm tip diameter) containing AAV1.hSyn.*Cre* (2E13 GC/ml), AAV1.CAG.Flex.*GCaMP6f* (1.3E13 GC/ml), and 50 μM FITC in PBS at a ratio of 1:1:1 were used to specifically target PNs and visually control the position and size of the viral injection, while <0.1 PSI pressure was used to inject the virus for 1 min. After virus injection the pipette was retracted, and the mouse was returned to its cage.

## Single neuron labeling with ANNINE-6plus

One week after virus injection, GCaMP6f-expressing PNs were targeted for voltage-sensitive dye (ANNINE-6plus) single-cell labeling by electroporation, guided by two-photon microscopy (*Kuhn and Roome, 2019*; *Roome and Kuhn, 2019*). GCaMP6f-expressing PN was electroporated using a patch pipette containing 3 mM ANNINE-6plus dissolved in ethanol. Borosilicate glass (patch) pipettes with 1 μm tip diameter (7–10 MΩ) were used for electroporation and a stimulus protocol of 50 negative current pulses (−30 μA), 1 ms in duration at 100 Hz were delivered. Neutral pressure was applied to the patch pipette to prevent leakage of the dye/ethanol solution into tissue and the pipette was retracted immediately after the cell was loaded and replaced for further single-neuron labeling.

Typically, two to five PNs were filled per mouse on the same day, and after loading PNs with dye, mice were returned to their cages to allow the dye to spread to distal dendrites and throughout the entire cell. ANNINE-6plus is highly lipophilic so dye diffusion can take several hours (>12 hr). After ~20 hr, the brightest labeled cells were selected for imaging experiments, which were performed the day following PN labeling. This also guaranteed that the PN was healthy and had not been damaged by the labeling procedure. Where possible, several PNs were labeled in the same mouse (up to 5) and used for simultaneous dendritic voltage and calcium imaging. Between neuron labeling sessions and following surgery, mice were returned to their cages and allowed to recover.

## Simultaneous dendritic voltage and calcium imaging

Approximately 20 hr after ANNINE-6plus labeling, simultaneous imaging from PN dendrites was performed in linescan mode at 2 kHz sampling rate. Labeled PNs were clearly visible in both red and green channels indicating successful labeling with ANNINE-6plus and GCaMP6f. During recordings, mice were alert and headfixed sitting on a rotating treadmill. Mice were allowed to sit awake on the treadmill for at least 1 hr before beginning the experiment. Bidirectional linescans, 512 pixels in width, lasting 10.5 s were performed at a line rate of 2 kHz. To limit photo-damage during linescans and to improve signal-to-noise ratio, the objective collar was rotated to elongate the excitation volume predominantly in the z-direction to ~5 μm. During the 10.5 s, no bleaching was observed. Fine corrections in linescan orientation (with respect to PN dendrites) were done prior to the experiment using the rotating stage, on which the mouse treadmill and micro-manipulators were placed. The linescans measuring 256 μm in width were carefully positioned as superficial as possible as to include the full dendritic width of the most distal dendritic spiny PN branches, thus maximizing the total membrane area covered by the linescan, typically less than 50 μm below the pia mater.

ANNINE-6plus is purely electrochromic, showing linear responses across the full physiological voltage range and is well suited for recording neuronal membrane potential, with a temporal resolution limited only by the fluorescence lifetime (*Fromherz et al., 2008*). The femtosecond-pulsed Ti: sapphire laser was used to excite fluorescence at 1020 nm, near the red spectral edge of absorption. To confirm optimal ANNINE-6plus sensitivity near the red spectral edge of absorption and the mechanism of voltage sensitivity (*Kuhn et al., 2004*), different excitation wavelengths were tested (*Kuhn and Roome, 2019*; *Roome and Kuhn, 2018*; *Roome and Kuhn, 2019*). Excitation near the red spectral edge of absorption to optimize voltage sensitivity allows for long-term simultaneous voltage and calcium dendritic recordings at least 500 s per recording session at different dendritic depths. As ANNINE-6plus is relatively hydrophobic compared to other voltage-sensitive dyes for intracellular application, the labeling lasts for at least 2 weeks (*Roome and Kuhn, 2018*; *Roome and Kuhn, 2019*; *Roome and Kuhn, 2020*). Due to an extended excitation point spread function (~1 × 1 × 5 μm$^3$) used to increase the signal-to-noise ratio, the voltage signal is the average membrane potential in this volume encompassing spines and dendritic shafts.

## Pharmacology and electrical stimulation of GCs

A micropipette inserted through the chronic cranial window access port was used for pharmacological manipulation. In this case, the micropipette: a beveled quartz micropipette (0.7 mm ID, Sutter Instruments) was placed ~50 μm below the PN soma for drug injections. Lidocaine (0.2%) (Sigma) and CNQX disodium salt (100 μM) (Tocris) were used to block Na$^+$ channels and AMPA receptors (excitatory synaptic input) respectively. AIDA (100 μM) (Tocris) was used to block mGluR1 activity. All drugs were dissolved in saline and applied by pressure injection (<0.5 psi) for 10 min prior to imaging and reduced to <0.1 psi while dendritic voltage and calcium recordings were repeated in the awake mice. Although it is not possible to confirm the exact concentration of drug applied in vivo, this procedure delivered a volume of solution in the nl range which spreads homogenously throughout the tissue and remained relatively localized (250–500 μm$^3$) to the injection area. This was confirmed by using the same delivery procedure to inject 50 μM FITC dissolved in saline, while observing the injection under the two-photon microscope (*Figure 1—figure supplement 1*). Because the exact concentration of drug is unknown in vivo, the degree of effect will be variable depending on the drug (binding constant and concentration). Dendritic recordings were occasionally repeated ~24 hr after drug application (guaranteeing drug washout) to confirm that the labeled PN was not physically damaged by the drug application. An example of drug washout is shown in *Figure 2—figure supplement 7*.

For electrical stimulation of GCs, a micropipette of the same type and dimensions used for drug application was filled with saline and positioned in the GC layer, 50–100 μm below the PN layer and 50–100 μm lateral to the labeled PN. A Master nine pulse generator and ISO-flex (A.M.P.I) was used to deliver current pulses to the GCs and evoke PF synaptic input onto PN dendrites (10 negative current stimuli, either 30 mA or 300 mA in amplitude, 1 ms duration, and at 100 Hz). In one recording, CNQX (100 μM) was also added to the stimulation micropipette saline, which was applied with pressure injection to block the electrical stimulus-evoked synaptic input (*Figure 1j*).

## Data analysis

Linescan TIFF images were initially cropped in width using ImageJ (US National Institute of Health) to contain only the dendrite from the labeled PN using the red channel as a guide, and to eliminate green signals originating from neighboring PNs. Full linescan traces were imported into Matlab and interpolated (from 2 to 10 kHz). All subsequent data analysis was performed using custom-written programs that we wrote previously with Matlab (MathWorks). Minor movement corrections were made to linescan images in the spatial dimension, to minimize movement artifacts that occur during the sensory stimulus. This was achieved by maximizing the cross correlation between the average pre-stimulus linescan spatial profile and the spatial profile of individual lines collected during the stimulus. It should be stated that because of the nature of linescan imaging, perfect movement correction cannot be achieved.

Single unit electrophysiological recordings were analyzed using Spike2 spike detection software (CED) to create a binary trace for simple spikes (SS) (and to eliminate spikes originating from neighboring PNs). All complex spikes (occurring at ~1 Hz) were easily identified by eye, which could be

confirmed by comparing simultaneous electrophysiology, voltage, and calcium traces. Complex spike binary traces were created using the initial sodium spike of the somatic complex spike to mark its onset. All binary traces, raw images, and electrophysiology were then imported into Matlab for analysis.

To calculate ΔF/F of full linescans, the green channel image was first scaled (using Matlab 'regress' function) and fit to the red channel and then subtracted from the red channel. This removed crosstalk from the green channel. Average baseline red channel fluorescence was then subtracted from the time-varying red channel fluorescence and the result was divided by the average baseline of red channel fluorescence to give ΔF/F for the red channel. The ΔF/F calculation for the green channel was made in the same way as for the red channel, but it was not necessary to first subtract red channel fluorescence from the green channel as the voltage signal was neglectable compared to GCaMP6f. Relative fluorescence changes imaged with an excitation wavelength of 1020 nm were converted with a factor of 2.1 mV/% to estimate voltage changes.

### DCS and DS spikelet detection in Matlab

DCS and single DS signals were detected using a custom-written program in Matlab. Spatially averaged voltage and calcium traces were first temporally filtered. Voltage traces were filtered using a 1 ms boxcar filter, while calcium traces were filtered using a 10 ms boxcar filter. All spikelets were then detected in the voltage traces using the 'findpeaks' function in Matlab, for which 'MinPeakHeight' was set at three standard deviations of the voltage recording, the 'MinPeakDistance' was set at 2.5 ms, and 'MinPeakProminence' was set at one standard deviation of the voltage recording. Spikelets were then sorted into DS or DCS signals based on the following criteria:

Spikelets were classed as DS signals if:

1. No additional spikelets were detected in a window of ±10 ms prior to and following the spikelet (i.e., no burst of spikelets was detected).
2. The average corresponding calcium signal measured in a 20 ms window following the spikelet increased by >1%, compared to the average calcium signal measured in a 20 ms window prior to the spikelet (i.e., corresponding calcium signal was detected).
3. The average voltage signal measured in a 10 ms window following the spikelet was less than one standard deviation of the voltage recording (i.e., the voltage recording returned immediately to baseline after a single spikelet).

Spikelets were classed as DCS signals if:

1. Additional spikelets were detected in a window of ±10 ms prior to and following the spikelet (i.e., a burst of two or more spikelets was detected).
2. The average corresponding calcium signal measured in a 20 ms window following the spikelet increased by >1% ΔF/F, compared to the average calcium signal measured in a 20 ms window prior to the spikelet (i.e., corresponding calcium signal was detected).

Classifications of DCS and DS signals were subsequently cross checked by visual inspection and through application of the classification algorithm, to paired dendritic and somatic recordings made on 7 PNs in awake mice (*Roome and Kuhn, 2018*), to confirm that DCS signals have a corresponding CF-evoked somatic complex spike, while DS signals had no corresponding somatic signal (i.e. purely dendritic signal) (see *Figure 1*). Where necessary, dendritic spikelets were removed from the voltage recording, by replacing voltage data in a window ±5 ms centered on each spikelet with the average voltage measurement 5 ms prior to the spikelet window, and the voltage data were subsequently filtered using a 10 ms boxcar filter.

### Behavior analysis

Behavioral recordings were made while the mouse was awake and sitting on a treadmill. A behavior camera (sCMOS), capturing frames at 100 fps, was used to record sensory stimulus-evoked movements, synchronized with the imaging. Linescan recordings (10.5 s in length) included a 100 ms sensory stimulus beginning after 5 s of recording, giving an air puff directed at the ipsilateral eye of the mouse. Air puffs (pressure: 30 psi) were delivered via a (0.7 mm I.D) glass capillary positioned 2 cm from the eye to evoke a reliable eye blink reflex in the ipsilateral eye. Regions of interest (ROIs) were drawn around the ipsilateral eye and the Δintensity/intensity was used to quantify eyelid position and closure during the stimulus, and traces were normalized such that 1 and 0 correspond to the

maximum and minimum eyelid closure recorded for each mouse across all recordings, in order to monitor changes in the eye response over time. Note that in some recordings a brief widening of the eye occurs immediately as the air puff is delivered (seen as a sharp positive deflection in the eyelid response traces). This is caused by the air puff blowing on the eyelid and causing it to widen momentarily, prior to the eye blink response.

### Data and software availability

Matlab codes are available at: https://github.com/cjroome/Roome_and_Kuhn_2020. Copy archived at swh:1:rev:25e88c9720e613065d56727e1225aa04fbf67e14.

Data is available at: https://doi.org/10.5061/dryad.6hdr7sqzt.

## Acknowledgements

We thank the GENIE Program and the Janelia Research Campus for distributing GCaMP6f, and Lina Koronfel and Soumen Jana for helpful feedback on the manuscript. We are grateful for generous support and funding from the Okinawa Institute of Science and Technology Graduate University.

## Additional information

### Funding

| Funder | Author |
| --- | --- |
| Okinawa Institute of Science and Technology Graduate University | Christopher J Roome<br>Bernd Kuhn |

The funders had no role in study design, data collection and interpretation, or the decision to submit the work for publication.

### Author contributions

Christopher J Roome, Conceptualization, Data curation, Software, Formal analysis, Validation, Investigation, Visualization, Methodology, Writing - original draft, Project administration, Writing - review and editing; Bernd Kuhn, Conceptualization, Supervision, Funding acquisition, Validation, Methodology, Project administration, Writing - review and editing

### Author ORCIDs

Christopher J Roome (ID) https://orcid.org/0000-0001-8936-668X

Bernd Kuhn (ID) https://orcid.org/0000-0002-6852-2433

### Ethics

Animal experimentation: All animal procedures were conducted in accordance with guidelines of the Okinawa Institute of Science and Technology Institutional Animal Care and Use Committee in an Association for Assessment and Accreditation of Laboratory Animal Care (AAALAC International)-accredited facility, under protocol numbers: 2016-170 and 2019-279.

### Decision letter and Author response

Decision letter https://doi.org/10.7554/eLife.59619.sa1

Author response https://doi.org/10.7554/eLife.59619.sa2

## Additional files

### Supplementary files

- Transparent reporting form

## Data availability

Matlab codes are available at: https://github.com/cjroome/Roome_and_Kuhn_2020 (copy archived at https://archive.softwareheritage.org/swh:1:rev:25e88c9720e613065d56727e1225aa04fbf67e14/) Data are available at: https://doi.org/10.5061/dryad.6hdr7sqzt.

The following dataset was generated:

| Author(s) | Year | Dataset title | Dataset URL | Database and Identifier |
|---|---|---|---|---|
| Roome CJ, Kuhn B | 2020 | Dendritic coincidence detection in Purkinje neurons of awake mice | https://doi.org/10.5061/dryad.6hdr7sqzt | Dryad Digital Repository, 10.5061/dryad.6hdr7sqzt |

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
