## [Decision Letter]

**Acceptance summary:**

Coincidence detection through dendritic integration in active dendrites has been proposed as an important mechanism for the regulation of synaptic strength. However, due to technical limitations, this issue has mainly been investigated in brain slices. Here the authors performed simultaneous high-speed voltage and calcium imaging in vivo to investigate dendritic integration in response to sensory input in cerebellar Purkinje cells. The paper is a technical tour de force. The main discovery, of interactions between parallel fiber and climbing fiber / complex spike signals in Purkinje cell dendrites, is an important contribution both to cerebellar physiology and to our understanding of dendritic integration and coincidence detection more broadly.

**Decision letter after peer review:**

Thank you for submitting your article "Dendritic coincidence detection in Purkinje neurons of awake mice" for consideration by *eLife*. Your article has been reviewed by Ronald Calabrese as the Senior Editor, a Reviewing Editor, and three reviewers. The following individual involved in review of your submission has agreed to reveal their identity: Samuel S-H Wang (Reviewer #1).

The reviewers have discussed the reviews with one another and the Reviewing Editor has drafted this decision to help you prepare a revised submission.

Summary:

The authors use high-speed voltage/calcium imaging paired with sensory stimulation to investigate dendritic integration in vivo in cerebellar Purknje cells. The reviewers were in agreement as to the high quality of the data presented from these technically challenging experiments, and agreed that the results, which provide confirmation of the in-vivo relevance of some well-known brain-slice results (Konnerth/Eilers/Augustine, Wang/Denk/Hausser,.…) were interesting and important.

However, the reviewers and editor all strongly agreed that the paper is not written clearly enough to convey either the essential findings, or to adequately place them within the context of previous work. Even for these expert reviewers, the paper was a challenging read. One reviewer spoke for us all in saying, "I was never sure that I understood the motivation for experiments or the logic in interpreting results." A major rewrite and changes to figures is necessary to make this paper interpretable, let alone suitable for the broad *eLife* audience. Because of the extensive and detailed comments of all three reviewers, their full reviews are appended below, with some of the most substantial concerns, and additional points that were raised during consultation, highlighted here.

Essential revisions:

Presentation:

Although mainly highlighted by reviewer 2, all reviewers agreed that the data need to be much better contextualized within the scope of the existing literature. Even the significant technical innovations were obscured by the presentation. Better context needs to be provided in the Introduction and Discussion, including more specific and direct references to relevant existing studies. Moreover, please explicitly state the logic, motivation, and interpretation, for each experiment and Figure within the Results section. In particular, as raised by multiple reviewers, the current results stand in contrast to papers by Gaffield and Christie, and this discrepancy needs to be directly addressed. See the comments of reviewer 2 for further specific guidance on presentation.

Definition/ quantification of supra and subthreshold:

The "coincidence/integration/threshold" language used throughout seemed vague or overly lofty, and failed to anchor the reader in the questions being addressed. Tying the current results more concretely to existing papers, including work by Wang, Hausser, Medina, Christie, Hull, and others, may help with this problem.

The general definition of sub and suprathreshold signals was confusing. As pointed out in reviewer 2 point 3, the observation of suprathreshold signals without detectable subthreshold signals is non-intuitive and requires more explanation. The authors seem to presume that this indicates distinct / independent origins in the two cases, but as pointed out by this comment and reviewer 3 point 1, this assumption needs to be directly addressed.

Relatedly, the manuscript would be strengthened by increased presentation of raw traces and better descriptions of the methods used to determine subthreshold voltage changes. The suggestion of reviewer 1 point 4, to show a histogram of the amplitudes of all single-trial voltage responses, could also help.

The authors are encouraged to pay close attention to improving the definition, explanation, and interpretation of the subthreshold/ suprathreshold language and presentation throughout the manuscript and the Figures.

Pharmacology experiments:

All three reviewers raised questions about the pharmacology experiments. In particular the rationale for analyzing the role of GABA-B rather than GABA-A was not made explicit in the main text, but it needs to be. In addition to the comments on pharmacology in the full reviews below, during consultation reviewer 1 made the following specific suggestion:

Considering the major role of inhibition in shaping Purkinje cell dendritic excitability, the authors should make some report of this in the main text. In particular, report the rate of spontaneous complex-spike-like events under unperturbed conditions (mean +/- SD and range) and compare it to the frequency events that they observed in the presence of the GABA_A blocker SR95531. This may also help in comparisons with Gaffield et al.

Please also address the specific points of each reviewer, below.

Reviewer #1:

Dendritic coincidence detection in Purkinje neurons of awake mice

The authors use recently developed methods from their lab for voltage and calcium imaging to study the mechanisms of calcium signal amplification in Purkinje neuron dendrites. This work nicely builds on their impressive previous work, recently published. Now the authors aim to identify mechanisms underlining coincidence detection in vivo during sensory-evoked responses.

The main discovery, of interactions between subthreshold (parallel fiber) and suprathreshold (climbing fiber / complex spike) signals in Purkinje cell dendrites, is an important contribution to cerebellar physiology. It directly confirms the biological relevance of signals discovered several decades ago in brain slices. The experiments seem to be competently performed, and the manuscript is well written.

We have some concerns, which can mostly be handled by additional analysis.

1) The authors address motion artifact, but given that small fluorescence fluctuations are central to the manuscript's claims, extra precaution is warranted. Can the authors demonstrate that animal motion does not contaminate the fluorescence traces, particularly in those as shown in Figure 2—figure supplement 1, Figure 1, Figure 2—figure supplement 2?

2) The authors' finding of in vivo supralinearity in Ca^2+^ signals driven by coincidence of CF and PF inputs stands in marked contrast to the findings of Gaffield et al., 2018 and 2019 as cited by the authors. Please include in the discussion a section to explain how such discrepant findings are possible.

3) Figure 1G: are these averages done trial-by-trial (i.e. sorting all depolarizing responses and then averaging them across cells), or cell-by-cell (i.e. averaging all trials for a single cell, then defining it as depolarizing, etc.)? One reason for asking is that in Figure 1H seems to show response properties of individual neurons. It would be better to label the four graphs in Figure 1H according to what they are – maybe each joined line segment is one cell, defined by its average response?

4) For depolarizing and hyperpolarizing responses, it would be helpful to show a histogram of the amplitudes of all single-trial voltage responses, to assess whether the responses are really categorical, as implied by the classification system – or whether the distribution is actually more continuous. This can be done for single cells or grouping all cells.

5) Subthreshold events are presumably driven by spatially clustered PF inputs. Is there evidence that the subthreshold calcium events are evoked across subcompartments of the imaged dendritic arbor? For example, through spatial scans from hotspots of response. It would be interesting to include a supplementary analysis addressing this issue.

6) Subsection “Sensory stimulation evokes sub- and suprathreshold signals in Purkinje neuron dendrites of awake mice”: calculate time to peak for these responses, and compare to the suprathreshold responses, as an additional test for a calcium release event. Also calculate t_1/2 of falling signal and compare.

7) Subsection “Sensory stimulation evokes sub- and suprathreshold signals in Purkinje neuron dendrites of awake mice”: for depolarizing signals, also give the time to maximum signal after stimulus **offset**. Inspection of the traces in Figure 1f suggests to me that the falling time courses of spontaneous and evoked signals have similar time courses when aligned to peak, which suggests the possibility that the evoked signals consist of the summed effect of continuous activation of a subthreshold signaling pathway (for example, perhaps having to do with motor output).

8) Figure 3E: I would have expected there to be some metabotropic signal (unless the voltage response is important for driving it). Please quantify the calcium signal at the time that the peak signal was observed in Figure 1I, and compare.

9) Subsection “Sensory stimulation evokes sub- and suprathreshold signals in Purkinje neuron dendrites of awake mice”: Explain why GABA_B mechanisms were tested but not GABA_A mechanisms. That would be an obvious source of hyperpolarizing effects. Even if you have only a few GABA_A experiments, it would be good to report them here.

10) As per *eLife* data-sharing policies, make simultaneous dendritic voltage and calcium imaging, paired extracellular somatic recordings available to catalyze the development of new algorithms for estimating spike rates from noisy calcium for Purkinje neurons.

11) It appears that animal subjects consistently widen their eye at onset of the puff stimulus. This is an unexpected result if the animals have no way to predict the stimulus. Can the authors explain this effect? Note that such widening does happen during eyeblink conditioning upon presentation of a novel conditional stimulus such as a tone or light flash. Perhaps there is some click or other salient cue that precedes the puff?

Reviewer #2:

This manuscript describes voltage-imaging and Ca-imaging recordings from Purkinje cell dendrites in mice subjected to air puffs to the eye that trigger eyelid closure. Correlations between the voltage and Ca signals are examined, and the effects of pharmacological agents are assessed, including blockade of AMPA receptors, Na channels, GABA-B receptors, and mGluR1 receptors. The measurements appear sophisticated, but the manuscript is difficult to read. It is hard to know what question is actually under investigation or what will be learned from the measurements. The authors bring up dendritic integration and coincidence detection, which are clearly fundamental to neural coding, but these are such generic terms that in the absence of any context their meaning becomes unclear. It is evident that the authors are interested in the supralinearity associated with voltage-gated conductances, especially Ca conductances, presumably because of the possibility of combinations of stimuli bringing Ca above a threshold to trigger a specific process, but none of this is actually stated or discussed explicitly in the manuscript, nor is it clear how the results bear on this question. There are certainly ongoing discussions in the field about conditions under which climbing fiber stimuli elicit greater or lesser Ca changes in Purkinje dendrites, and some of these papers are cited obliquely but the text does not help the reader understand the controversy, its significance/importance, or how the present results address any aspects of it. Overall, the experiments lack both rationale and interpretation. As such, they come across as a string of observations rather than a coherent study.

1) The Introduction and Discussion do not provide a clear scientific context for the study. No real question is posed in the introduction, and no in-depth connection to other studies or ideas is offered in the discussion. Even the value of the technical innovations is not explained in a way that makes a reader appreciate their possible significance. As a consequence, the work reads like a report of a series of measurements, but it is hard to know what is learned, except that if there are multiple sources of depolarization, voltage-gated channels enter their typical positive feedback loops and increase the total Ca influx, which stands to reason.

2) One of the relatively novel aspects of the study is the voltage imaging in conjunction with Ca imaging (which is theoretically valuable), but the manuscript lacks description of what the voltage signals represent. It is stated that the imaging rate is 2 KHz, but it is not clear what the sensitivity, resolution, or calibration might be, in a way that lets the reader know how the deflections relate to action potentials or voltage responses of the cells. What is the smallest voltage change detectable? Is there filtering/distortion and how does it affect the results? Can one read the deflections like an electrophysiological recording? Some of this information might be implicit in Figure 1—figure Supplement 1B, but it is not a focus or even commented on in the main text, as far as I can tell.

3) Subsection “Sensory stimulation evokes sub- and suprathreshold signals in Purkinje neuron dendrites of awake mice” and beyond. The authors observe suprathreshold signals without detectable subthreshold signals. This idea is non-intuitive and requires some more explanation, or at least should be returned to later on, which as far as I can tell it is not. Presumably such an instance indicates that the supra and subthreshold signals can be completely distinct / independent in their origins, but if so, it raises the question of the conditions that separate and the conditions that link them. While the authors need not figure everything out, they should at least offer some commentary on this point. It also seems at odds with the voltage-calcium relationship that they present at the end of Figure 1.

4) In the pharmacology studies, it is not clear what kind of criteria the authors used to determine how much drug reached how large of an area. Was drug just applied until there was an effect? What was the point of the lidocaine experiments? It does not seem surprising that blocking either voltage-gated channels or AMPA receptors would largely suppress signals; were these meant as controls for comparing the CGP and AIDA experiments or was there some question as to whether certain effects might actually be insensitive to blockade of Na channel or fast glutamatergic transmission? Also, why were GABA-B receptors chosen (as opposed to GABA-A receptors, or as opposed to other metabotropic receptors)? The experiments require a better rationale.

5) I think the main discovery is intended to be Figure 6F in which the Ca signal is enhanced when dendritic spikes, presumably from PF activity, follow CF activity, and then Figure 7 makes the case that mGluR1 receptors contribute to the supralinearity. It might be useful to anticipate this link and/or bring up a little more about what is known about mGluR1 signaling in Purkinje cells in both intro and discussion.

Reviewer #3:

The authors use in vivo two-photon imaging to simultaneously monitor calcium and membrane potential in cerebellar PN dendrites. The sophisticated techniques have been carefully developed by the investigators. The resulting data is compelling, and could ultimately clarify how PN dendrites integrate inputs to modulate postsynaptic activity and synaptic plasticity. The central finding here is that sensory-evoked input drives sub- and supra-threshold signals in PN dendrites. Coincidence of these signals (which is interpreted as originating from PFs and CFs, respectively) produces supralinear calcium responses in PNs. Despite the beauty of the data, and the clear utility of these techniques to investigate the mechanisms of motor learning in the cerebellum, the interpretation of the synaptic source of dendritic signals (particularly PFs) goes beyond the data. However, this central findings are still of great interest to the field.

1) Based on their previous work the authors state definitively that DS signals are the result of PF input (Roome and Kuhn, 2018). However, they previously "predicted" that PFs drove DS signals. Subthreshold dendritic depolarizations are similarly inferred to originate from PF input without evidence (Subsection “Sensory stimulation evokes sub- and suprathreshold signals in Purkinje neuron dendrites of awake mice”). While DS and subthreshold signals are very likely to originate from PF input, language throughout the manuscript could easily be amended to reflect the fact that this is a conjecture. Alternatively, new experiments could be used to manipulate granule cell activity (optogenetics, DREADDs) or selectively inhibit transmission from PFs (for example, release from PFs, but not CFs, is blocked by the group III mGluR agonist L-AP; Hashimoto and Kano, 1998; Kreitzer and Regehr, 2001). Attributing DCS signals to CFs is more clearly justified.

2) While the mechanism of DS signals might be over-interpreted, the mechanisms that motivated pharmacological experiments are barely discussed, leaving the reviewer uncertain as to why these experiments are included. For example, GABAB receptors are noted as present at the PF-PN synapse but their localization and function are not discussed, no prediction is made about the effect of blockade, and these experiments are not discussed outside the Results section. In a manuscript full of excellent data, these experiments should be omitted or explained in greater detail.

[Editors' note: further revisions were suggested prior to acceptance, as described below.]

Thank you for resubmitting your work entitled "Dendritic coincidence detection in Purkinje neurons of awake mice" for further consideration by *eLife*. Your revised article has been evaluated by Ronald Calabrese (Senior Editor) and a Reviewing Editor, and two peer reviewers.

The manuscript has been improved but there are some remaining issues that need to be addressed before acceptance, as outlined below:*Reviewer 1:*

This manuscript is very improved since the previous submission. It is a thorough and well-done piece of work exploring phenomena that in years past were considered to be near-impossible to investigate, namely the local dendritic integration of coincident synaptic signals. Congratulations are due to the authors for carrying out this technically challenging and biologically convincing study.

Reviewer 3:

The revised manuscript from Roome and Kuhn addresses many of the concerns raised by reviewers. Most important, they substantially rewrote the introduction and Discussion sections to provide better motivation for this study, and comparison to past results. Pharmacology experiments that were not central to their findings and distracted from the main argument have been removed. They performed additional experiments using electrical stimulation evoke PF input to PNs, and the results lend credence to the idea that DS signals originate from PF activity, which is central to all of their interpretations. The conclusions they arrive at from these technically impressive experiments, and how these conclusions contrast with previous interpretations, are now more clearly stated in the text. The manuscript is now better-suited for publication in e*Life*.

1) Subsection “Dendritic spikes triggered by coincident CF and PF synaptic input evoke supralinear calcium signals during sensory stimulation”: Is the point here that under single-DCS conditions, the voltage traces do not show signs of voltage-based amplification, yet the calcium signals still show enhancement? And is the additional source of calcium potentially a source of calcium release or influx that is not voltage-dependent? If so, it would help the reader if you ended the paragraph with an explanatory statement.

2) Subsection “Dendritic spikes triggered by coincident CF and PF synaptic input evoke supralinear calcium signals during sensory stimulation”, supralinearity of calcium signals: Have you tried doing the same analysis but with the voltage signals?

3) Subsection “Dendritic spikes triggered by coincident CF and PF synaptic input evoke supralinear calcium signals during sensory stimulation”, Figure 4H/I: The timing-dependence curve here shows peak calcium signals at DCS preceding DS by 0-80 ms. If the DCS were a CF-evoked complex spike only, this would seem to be unexpected – it is opposite to timing-dependence measured in brain slices. Since the graph is made in the same style as a similar test in Wang/Denk/Hausser, the discrepancy comes to mind. It could be tied into the next paragraph, where you then explain that the DCS isn't just a complex spike, but has a pre-depolarization. (Can you report near here in the text the duration of the pre-DCS voltage response for comparison?) Also, the DS comes as the culmination of integration of PF input. Again, discussion of the duration of that signal would be helpful. Generally, a critical comparison with the Wang et al0., PF-before-CF timing-dependence finding would be useful. I think the two findings are consistent, but it needs explaining.

4) Figure 4H: In the 0-80 ms timing window, the relative calcium peak appears to be largest for 2 or 3 spikelets, and smaller for 4 or 5 spikelets. That is unexpected. Or maybe it is just hard to read the graph. Report statistics on that please.

5) To address the issue of motion artifacts (reviewer #1 comment 1) the authors suggest that A) the different time-course of voltage and calcium signals argues against motion artifacts underlying these signals, and B) the ability to block both signals with CNQX also suggests synaptic origin. They further state that some analysis techniques were employed to compensate for motion (Subsection “Data analysis”) and mention in the rebuttal that trials were omitted if the mouse made a large movement. Neither of these methods seems to conclusively rule out movement artifacts. This caveat should be mentioned more explicitly in the manuscript.

6) In general, the textual revisions have greatly improved the readability and logical flow of the manuscript. However, this sentence in the abstract is still confusing and should be rewritten to better align with their new presentation of the data: "Sensory stimulation evokes post-synaptic potentials that coincide with climbing fiber evoked dendritic complex spikes and parallel fiber evoked dendritic spikes." Where do the spikes come from if not from post-synaptic potentials? Possible solutions would be to replace "coincide with" to "drive", or something like "Sensory stimulation increases the rate of postsynaptic potentials and dendritic calcium spikes evoked by both climbing fiber and parallel fibers." The rewritten statement should also be sure to clarify the "these" in the subsequent sentence.

7) Since the previous submission, another paper has come out that the authors should mention alongside their treatment of the Gaffield and Christie paper in the Discussion: Roh et al., 2020

---

## [Author Response]

Essential revisions:Presentation:Although mainly highlighted by reviewer 2, all reviewers agreed that the data need to be much better contextualized within the scope of the existing literature. Even the significant technical innovations were obscured by the presentation. Better context needs to be provided in the Introduction and Discussion, including more specific and direct references to relevant existing studies. Moreover, please explicitly state the logic, motivation, and interpretation, for each experiment and Figure within the Results section. In particular, as raised by multiple Reviewers, the current results stand in contrast to papers by Gaffield and Christie, and this discrepancy needs to be directly addressed. See the comments of reviewer 2 for further specific guidance on presentation.Definition/ quantification of supra and subthreshold:The "coincidence/integration/threshold" language used throughout seemed vague or overly lofty, and failed to anchor the reader in the questions being addressed. Tying the current results more concretely to existing papers, including work by Wang, Hausser, Medina, Christie, Hull, and others, may help with this problem.The general definition of sub and suprathreshold signals was confusing. As pointed out in reviewer 2 point 3, the observation of suprathreshold signals without detectable subthreshold signals is non-intuitive and requires more explanation. The authors seem to presume that this indicates distinct / independent origins in the two cases, but as pointed out by this comment and reviewer 3 point 1, this assumption needs to be directly addressed.Relatedly, the manuscript would be strengthened by increased presentation of raw traces and better descriptions of the methods used to determine subthreshold voltage changes. The suggestion of reviewer 1 point 4, to show a histogram of the amplitudes of all single-trial voltage responses, could also help.The authors are encouraged to pay close attention to improving the definition, explanation, and interpretation of the subthreshold/ suprathreshold language and presentation throughout the manuscript and the Figures.Pharmacology experiments:All three reviewers raised questions about the pharmacology experiments. In particular the rationale for analyzing the role of GABA-B rather than GABA-A was not made explicit in the main text, but it needs to be. In addition to the comments on pharmacology in the full reviews below, during consultation reviewer 1 made the following specific suggestion:Considering the major role of inhibition in shaping Purkinje cell dendritic excitability, the authors should make some report of this in the main text. In particular, report the rate of spontaneous complex-spike-like events under unperturbed conditions (mean +/- SD and range) and compare it to the frequency events that they observed in the presence of the GABA_A blocker SR95531. This may also help in comparisons with Gaffield et al.Please also address the specific points of each reviewer, below.

Thank you for your comments. To address the reviewers concerns we have made major revisions to the manuscript, including re-writing. We have performed additional experiments and present more data, including raw data, and the requested data analyses. We believe the manuscript reads much clearer now and fits better with the existing work. We appreciate the reviewer’s advice.

We removed the data involving inhibitory synaptic input because this distracted from the major focus (parallel fiber and climbing fiber input) and we agree with the reviewers that this aspect requires further experiments since GABA_A, GABA_B and also calcium- and voltage-activated potassium channels in PN dendrites potentially contribute to the hyperpolarizing signals that we see, and we do not yet know how they interact – we are currently exploring this in more detail and will address this in future work.

Reviewer #1:[…]1) The authors address motion artifact, but given that small fluorescence fluctuations are central to the manuscript's claims, extra precaution is warranted. Can the authors demonstrate that animal motion does not contaminate the fluorescence traces, particularly in those as shown in Figure 2—figure supplement 1, Figure 1, Figure 2—figure supplement 2?

We have now added the 2D line-scan data (Figure 2 and Figure 2 —figure supplement 3), each showing large depolarizing and hyperpolarizing voltage signals during the stimulus with no movement artifacts. We note that the voltage and calcium linescans show very different signals on different time scales during the stimulus. This is particularly true for the hyperpolarizing voltage signal. This would not be the case for a movement artefact which would lead to similar transients in both channels. Such transients in combination with movement detected by the behavior camera can be used to detect traces with potential movement artefacts. All traces were inspected carefully and trials with large mouse movements, such as strong flinching, movement of limbs or running, were not included.

We would also like to point out that the pharmacology experiments serve as a control for confirming the signals are synaptic and not due to movement artefacts.

2) The authors' finding of in vivo supralinearity in Ca^2+^ signals driven by coincidence of CF and PF inputs stands in marked contrast to the findings of Gaffield et al., 2018 and 2019 as cited by the authors. Please include in the discussion a section to explain how such discrepant findings are possible.

We have included comments in the Discussion. Gaffield et al., do not directly record PF input or voltage in the PN dendrites and so can’t unequivocally make this claim. The trial-by-trial averaging of their calcium imaging experiments is also problematic and is likely to average out the supralinear dendritic calcium responses (which are in reality highly variable) – essentially linearizing the CF-PN, input-output relationship.

3) Figure 1G: are these averages done trial-by-trial (i.e. sorting all depolarizing responses and then averaging them across cells), or cell-by-cell (i.e. averaging all trials for a single cell, then defining it as depolarizing, etc.)? One reason for asking is that in Figure 1H seems to show response properties of individual neurons. It would be better to label the four graphs in Figure 1H according to what they are – maybe each joined line segment is one cell, defined by its average response?

These traces are averages from a signal cell (top) and from the group (middle). We have now described this in the figure and caption more clearly.

4) For depolarizing and hyperpolarizing responses, it would be helpful to show a histogram of the amplitudes of all single-trial voltage responses, to assess whether the responses are really categorical, as implied by the classification system – or whether the distribution is actually more continuous. This can be done for single cells or grouping all cells.

We agree this was missing and have now included the histogram in Figure 2, showing the maximum change in dendritic voltage during the stimulus.

5) Subthreshold events are presumably driven by spatially clustered PF inputs. Is there evidence that the subthreshold calcium events are evoked across subcompartments of the imaged dendritic arbor? For example, through spatial scans from hotspots of response. It would be interesting to include a supplementary analysis addressing this issue.

Yes, this is an interesting point. We plan to carefully address the spatial pattern of PF input etc. in the near future. Here, the problem is that the one-dimensional imaging (line scanning) is not sufficient to map the responses properly. Therefore, we are currently building a new setup which will allow fast two-dimensional imaging to follow up on this issue.

6) Subsection “Sensory stimulation evokes sub- and suprathreshold signals in Purkinje neuron dendrites of awake mice”: calculate time to peak for these responses, and compare to the suprathreshold responses, as an additional test for a calcium release event. Also calculate t_1/2 of falling signal and compare.

We have now included this analysis on all calcium signals, and we show now these signals in Figure 4—figure supplement 1 for a clearer comparison of their scale and time course.

7) Subsection “Sensory stimulation evokes sub- and suprathreshold signals in Purkinje neuron dendrites of awake mice”: for depolarizing signals, also give the time to maximum signal after stimulus **offset**. Inspection of the traces in Figure 1f suggests to me that the falling time courses of spontaneous and evoked signals have similar time courses when aligned to peak, which suggests the possibility that the evoked signals consist of the summed effect of continuous activation of a subthreshold signaling pathway (for example, perhaps having to do with motor output).

We added the following to subsection “Sensory stimulation evokes graded depolarizing and hyperpolarizing signals and dendritic spikes in Purkinje neuron dendrites”, “Depolarizing signals increased gradually throughout the stimulus, beginning 9 ± 3 ms (mean ± SD, n = 307) after stimulus onset, and reaching a maximum after 101 ± 3 ms, essentially at the stimulus offset”

In our previous paper we showed that a non-evoked EPSP lasts for about 10 ms. As the post-stimulus voltage signal lasts for over 100 ms it is likely that it is caused by superposition of sustained PF input, as suggested by reviewer #1.

8) Figure 3E: I would have expected there to be some metabotropic signal (unless the voltage response is important for driving it). Please quantify the calcium signal at the time that the peak signal was observed in Figure 1I, and compare.

We have now included this analysis on all calcium signals, and we show now these signals in Figure 4—figure supplement 1 for a clearer comparison of their scale and time course.

9) Subsection “Sensory stimulation evokes sub- and suprathreshold signals in Purkinje neuron dendrites of awake mice”: Explain why GABA_B mechanisms were tested but not GABA_A mechanisms. That would be an obvious source of hyperpolarizing effects. Even if you have only a few GABA_A experiments, it would be good to report them here.

We removed the data involving inhibitory synaptic input because this distracted from the major focus (parallel fiber and climbing fiber input) and we agree with the reviewers that this aspect requires further experiments since GABA_A, GABA_B and also calcium activated potassium channels in PN dendrites potentially contribute to the hyperpolarizing signals. We do not yet know how they interact but we are currently exploring this in more detail and will address this in future work.

10) As per eLife data-sharing policies, make simultaneous dendritic voltage and calcium imaging, paired extracellular somatic recordings available to catalyze the development of new algorithms for estimating spike rates from noisy calcium for Purkinje neurons.

Yes, we are happy to do this.

11) It appears that animal subjects consistently widen their eye at onset of the puff stimulus. This is an unexpected result if the animals have no way to predict the stimulus. Can the authors explain this effect? Note that such widening does happen during eyeblink conditioning upon presentation of a novel conditional stimulus such as a tone or light flash. Perhaps there is some click or other salient cue that precedes the puff?

In these recordings the widening happens only immediately when the airpuff is delivered, briefly blowing on the eye lid and causing it to widen momentarily.

We have now added this to the Materials and methods section: “Note that in some recordings a brief widening of the eye occurs immediately as the air puff is delivered (seen as a sharp positive deflection in the eyelid response traces). This is an artefact caused by the air puff briefly blowing on the eyelid and causing it to widen momentarily, prior to the eye blink response.”

Reviewer #2:This manuscript describes voltage-imaging and Ca-imaging recordings from Purkinje cell dendrites in mice subjected to air puffs to the eye that trigger eyelid closure. Correlations between the voltage and Ca signals are examined, and the effects of pharmacological agents are assessed, including blockade of AMPA receptors, Na channels, GABA-B receptors, and mGluR1 receptors. The measurements appear sophisticated, but the manuscript is difficult to read. It is hard to know what question is actually under investigation or what will be learned from the measurements. The authors bring up dendritic integration and coincidence detection, which are clearly fundamental to neural coding, but these are such generic terms that in the absence of any context their meaning becomes unclear. It is evident that the authors are interested in the supralinearity associated with voltage-gated conductances, especially Ca conductances, presumably because of the possibility of combinations of stimuli bringing Ca above a threshold to trigger a specific process, but none of this is actually stated or discussed explicitly in the manuscript, nor is it clear how the results bear on this question. There are certainly ongoing discussions in the field about conditions under which climbing fiber stimuli elicit greater or lesser Ca changes in Purkinje dendrites, and some of these papers are cited obliquely but the text does not help the reader understand the controversy, its significance/importance, or how the present results address any aspects of it. Overall, the experiments lack both rationale and interpretation. As such, they come across as a string of observations rather than a coherent study.

We appreciate your comments. We have made significant changes to the manuscript to address your concerns. We believe the manuscript is now much easier to read, both rationale and interpretation of results are clearly stated for a more coherent study. Below are examples of where your concerns have been addressed.

– In the Introduction we introduce the importance for studying dendritic integration in awake animals and note how useful recent in vivo studies are: “Dendritic integration is fundamental to signal processing in the brain. So far, most studies on dendritic integration were performed in vitro, in the absence of physiological inputs (Larkum et al., 2009; Markram et al., 1997; Stuart and Häusser, 2001; Wang et al., 2000). Equivalent in vivo studies, though technically challenging, are essential for exploring how dendritic integration works in living animals (Chen et al., 2013; Jia et al., 2010; Murayama et al., 2009; Palmer et al., 2014; Sheffield and Dombeck, 2015; Smith et al., 2013).” And we note: “As such, our understanding of the basic components of dendritic integration, the frequency, amplitude, and spatiotemporal distribution of synaptic inputs under physiological conditions, and how these inputs are integrated by dendrites in awake behaving animals, remains incomplete.”

– In the Introduction we explain why using voltage imaging in awake animals is useful for investigating how dendritic integration works in vivo, and we highlight the limitations that current techniques face in investigating dendritic integration: “Conventional in vivo dendritic recording techniques have faced important challenges for studying dendritic integration. Notably, in vivo patch clamp recordings targeting the soma or smooth dendrites cannot directly measure signals in the most distal dendritic regions that receive the majority of synaptic input, called the “spiny” dendrites. in vivo calcium imaging from spiny dendrites can partially circumvent this issue (Chen et al., 2012). However, this approach measures dendritic calcium events only, and omits the depolarizing and hyperpolarizing voltage signals evoked by synaptic input that do not trigger postsynaptic calcium signals. These synaptic potentials are expected to generate continuous but highly inhomogeneous spatiotemporal dendritic activity in awake animals.” And in the Introduction we explain that “a direct relationship between voltage and calcium signalling in spiny dendrites has not yet been explored in behaving animals”. This is a particularly important novel aspect of this study because we can make a direct correspondence between synaptic input evoked dendritic spikes (voltage) and their resulting calcium signals – and we use this to describe the non-linear dendritic mechanisms (dendritic coincidence detection) during sensory stimulation.”

– In the Introduction we introduce a form of dendritic integration called dendritic coincidence detection and explain its relevance to PF and CF input onto cerebellar PN dendrites: “Coincidence detection is a basic form of dendritic integration. By detecting coincident synaptic input, it is thought that neurons distinguish important signals from ongoing synaptic activity and modify synaptic strength through synaptic plasticity…”.

– In the Introduction we give an overview of the most relevant study of dendritic coincidence detection in vitro, and introduce mGluR1s and voltage-dependant dendritic mechanisms: “Specifically, previous in vitro studies identified two dendritic mechanisms by which coincidence detection may occur, that depend on the strength and timing of coincident PF and CF stimulation (Wang et al., 2000). These mechanisms involved voltage-gated calcium channels and/or group 1 metabotropic glutamate receptors (mGluR1). How these processes occur during behavior remains to be determined.” Our study follows up directly on these observations made in vitro.

– In the Introduction we give an overview of most relevant studies in vivo, that (as pointed out by reviewer #1) make contradictory conclusions, and so we highlight the motivation for a more in-depth study into the dendritic coincidence detection process. “in vivo studies using calcium imaging found that climbing fiber evoked calcium signals were enhanced during sensory stimulation (an air puff directed towards the eye) in mice. Notably, graded non-climbing fiber sensory evoked calcium signals were detected and proposed to be driven by PF input (Najafi et al., 2014a; Najafi et al., 2014b). More recent studies however have failed to detect supralinear dendritic calcium signals triggered by coincident PF and CF input during sensory stimulation (Gaffield et al., 2019; Gaffield et al., 2018), challenging the role that PF input might play on the sensory evoked dendritic calcium signals.”

– In the Introduction we reiterate what is missing from the current literature and explain how our simultaneous voltage/calcium imaging technique allows us to address this: “Coincidence detection of PF and CF input remains to be confirmed in vivo. Specifically, it is unclear if the coincidence detection mechanisms previously described in vitro, also occur in vivo and if so, how they occur and under what behavioral conditions. Furthermore, because of technical limitations, a direct relationship between voltage and calcium signaling in spiny dendrites has not yet been explored in behaving animals.

Here by combining simultaneous voltage and calcium imaging from the PN spiny dendrites, we record dendritic calcium spikes and post-synaptic potentials to investigate sensory evoked dendritic integration and coincidence detection in PN dendrites of awake mice. Specifically, we investigate the dendritic mechanisms that lead to the enhanced dendritic calcium signals in PNs during sensory stimulation.”

– To help with interpretation of the results and introduce our approach more gradually we performed additional experiments (now Figure 1), whereby PFs were electrically stimulated and the corresponding dendritic voltage and calcium signals in PNs were described. This provides a more intuitive bridge between the in vitro experiments and ours. It also allows us to introduce the different voltage signals evoked by PF input (depolarizing/hyperpolarizing and DS) and supra-linear calcium signals more systematically. We end the figure with an example of the dendritic coincidence detection process, that we will be describing in the remaining figures (albeit non-physiological at this point).

– To aid in the interpretation of results and rationale for subsequent stages in the analysis we have now included results sub-headings and a short summary with relevant references for each figure, as follows:

Figure 1, Figure 2 and Figure 3 focus on PF input to PN dendrites primarily

Figure 1: “Electrical stimulation of PF input evokes graded dendritic voltage and calcium signals and dendritic spikes in Purkinje neuron dendrites.”

Subsection “Electrical stimulation of PF input evokes graded dendritic voltage and calcium signals and dendritic spikes in Purkinje neuron dendrites”: “Taken together this indicates that dendritic calcium signals evoked by PF synaptic input have multiple calcium sources, as described previously in vitro (Denk et al., 1995; Eilers et al., 1995; Llano et al., 1991a; Sugimori and Llinás, 1990; Wang et al., 2000). These are likely to include calcium influx through mGluR1s, which are expressed at PF-PN synapses (Tempia et al., 2001), voltage dependent dendritic spikes (Denk et al., 1995; Roome and Kuhn, 2018; Usowicz et al., 1992), and calcium released from internal stores (Llano et al., 1994; Llano et al., 1991a; Wang et al., 2000). Since PF and CF synaptic input constitute the only excitatory input to PN dendrites (Llano et al., 1991b), and DS events are blocked by CNQX (Figure 1J see also Figure 2 – figure supplement 7), have no corresponding somatic complex spike associated with CF input (Figure 1H), and are evoked by strong PF stimulation (Figure 1I-K), we conclude that the DS events are generated by strong PF input to the PN dendrites. As a proof of principle for dendritic coincidence detection of PF and CF input, Figure 1K shows an example of how PF input (here evoked by electrical stimulation) following a spontaneous CF event triggers additional DS events and results in a supralinear dendritic calcium signal. While electrical stimulation of GCs may be unphysiological here, this demonstrates how supralinear dendritic calcium signals can be generated by coincident PF and CF input to PN dendrites in vivo.”

Figure 2: “Sensory stimulation evokes graded depolarizing and hyperpolarizing signals and dendritic spikes in Purkinje neuron dendrites.”

Subsection “Sensory stimulation evokes graded depolarizing and hyperpolarizing signals and dendritic spikes in Purkinje neuron dendrites”: From the observations made thus far we conclude that sensory stimulation evokes graded depolarizing and hyperpolarizing voltage signals in PN spiny dendrites. On average these dendritic voltage signals have a relatively small amplitude (6.1 ± 9.0 mV), below threshold for triggering voltage gated calcium channels, have a much slower time course compared to dendritic spikes and are blocked by CNQX (Figure 2K and Figure 2—figure supplement 7). Since CF synaptic input reliably evokes rapid dendritic spikes (Roome and Kuhn, 2018) and noting their similarity to the voltage signals evoked by PF stimulation shown in Figure 1I-K, we conclude that these signals originate primarily from PF synaptic input. On average the sensory stimulus evoked PF input gives rise to a depolarizing signal across the PN dendrites (global depolarization) that begins 9 ± 3 ms after stimulus onset, which is consistent with rapid and dense GC responses to sensory stimulation (Giovannucci et al., 2017; Jorntell and Ekerot, 2006). The sensory stimulation evoked hyperpolarizing dendritic signals are also in agreement with an increase in MLI activity, and feed forward inhibition evoked by sensory stimulation (Chu et al., 2012; Gaffield et al., 2018). The trial by trial variability in PN dendritic voltage responses that we detect is likely to result from an interplay between these two synaptic inputs driven by the MF pathway.

Figure 3: “Sensory stimulation evokes graded voltage dependent dendritic calcium signals in absence of dendritic complex spikes.”

Subsection “Sensory stimulation evokes graded depolarizing and hyperpolarizing signals and dendritic spikes in Purkinje neuron dendrites”: “This confirms that PF evoked dendritic voltage signals (described in Figure 2) generate voltage dependent dendritic calcium signals in absence of CF input and dendritic spikes.”

Figure 4, Figure 5 focuses on the interaction of PF and CF input and in particular on evoked dendritic spikes and associated calcium signals.

Figure 4: “Dendritic spikes triggered by coincident CF and PF synaptic input evoke supralinear calcium signals during sensory stimulation.”

Subsection “Dendritic spikes triggered by coincident CF and PF synaptic input evoke supralinear calcium signals during sensory stimulation”: “Thus, we conclude that paired DCS + DS signals evoked by coincident CF and PF input act to enhance the dendritic calcium signal supralinearly, and this enhancement is dependent on the timing of paired DCS + DS events. However, paired DCS + DS signals are relatively rare (detected in only 5.9 ± 0.9 % of recordings), suggesting an additional coincidence detection mechanism at work. Taken together, our findings thus far reveal a predominantly depolarizing voltage signal and an increase in dendritic spike activity during sensory stimulation. Both these signals are blocked by CNQX and so originate primarily from coincident excitatory PF and CF synaptic input.”

Figure 5: “Supralinear dendritic signaling in PN dendrites during sensory evoked stimulation is voltage and mGluR1 dependent.”

Subsection “Supralinear dendritic signaling in PN dendrites during sensory evoked stimulation is voltage and mGluR1 dependent”: “indicating that the contribution of mGluR1 activity to the calcium signal is specific to sensory evoked calcium signals when CF and PF inputs coincide, and we propose that mGluR1s are activated at PF-PN synapses, where the PF and CF synaptic potentials converge.” And “This suggests that mGluR1 activity enhances spikelet generation during evoked DCS signals.”

– Similarly, we have now divided the Discussion into the following subheadings:

In subsection: “Dendritic coincidence detection mechanisms modulate sensory stimulation evoked dendritic calcium signals in Purkinje neuron dendrites”, we summarize the key findings and three forms of dendritic coincidence detection that we observe.

In subsection: “Functional implications: A solution to the “signal-to-noise” problem.” We discuss the broader functional implications of dendritic coincidence detection at PN dendrites in the cerebellum, and how it fits with the current literature. In particular we refer to two recent reviews (Najafi and Medina, 2013) and (Zang and De Schutter, 2019) that make predictions directly related to our findings.

1) The Introduction and Discussion do not provide a clear scientific context for the study. No real question is posed in the introduction, and no in-depth connection to other studies or ideas is offered in the discussion. Even the value of the technical innovations is not explained in a way that makes a reader appreciate their possible significance. As a consequence, the work reads like a report of a series of measurements, but it is hard to know what is learned, except that if there are multiple sources of depolarization, voltage-gated channels enter their typical positive feedback loops and increase the total Ca influx, which stands to reason.

We have made major revisions to the manuscript. We hope and believe the manuscript now reads better and the aims are clearer and fit better with the existing work. We appreciate the reviewer’s advice.

2) One of the relatively novel aspects of the study is the voltage imaging in conjunction with Ca imaging (which is theoretically valuable), but the manuscript lacks description of what the voltage signals represent. It is stated that the imaging rate is 2 KHz, but it is not clear what the sensitivity, resolution, or calibration might be, in a way that lets the reader know how the deflections relate to action potentials or voltage responses of the cells. What is the smallest voltage change detectable? Is there filtering/distortion and how does it affect the results? Can one read the deflections like an electrophysiological recording? Some of this information might be implicit in Figure 1—figure supplement 1B, but it is not a focus or even commented on in the main text, as far as I can tell.

We added the following sentences to help explain.

Subsection “Electrical stimulation of PF input evokes graded dendritic voltage and calcium signals and dendritic spikes in Purkinje neuron dendrites” “Two photon linescan imaging (2 kHz) recorded dendritic voltage and calcium signals simultaneously from the spiny PN dendrites, predominantly comprising PF-PN synapses, at a depth of 30-70 mm below the dura (Figure 1F). […] The relative fluorescence change of ANNINE-6plus can be used to estimate the average membrane voltage change in PNs with a conversion factor of 2.1 mV/% (Kuhn et al., 2004; Roome and Kuhn, 2018).”

Regarding the question for the smallest voltage change that can be detected: The voltage sensitive dye reports voltage changes linearly. However, the voltage signal has to overcome the inherent noise of optical measurements. In other words, the more photons can be collected, the smaller voltage changes can be detected. For example, in our previous paper (Roome and Kuhn, 2018) we showed backpropagating action potentials in PN which are about 1mV at a signal-to-noise ratio of about 10:1, but only after extensive averaging. For a detailed description of the method, see Kuhn and Roome 2019. For detailed protocols, see Roome and Kuhn 2019.

3) Subsection “Subsection “Sensory stimulation evokes sub- and suprathreshold signals in Purkinje neuron dendrites of awake mice” and beyond. The authors observe suprathreshold signals without detectable subthreshold signals. This idea is non-intuitive and requires some more explanation, or at least should be returned to later on, which as far as I can tell it is not. Presumably such an instance indicates that the supra and subthreshold signals can be completely distinct / independent in their origins, but if so, it raises the question of the conditions that separate and the conditions that link them. While the authors need not figure everything out, they should at least offer some commentary on this point. It also seems at odds with the voltage-calcium relationship that they present at the end of Figure 1.

For clarity we no longer use “subthreshold” and “suprathreshold”. Instead, we use “dendritic spikes” and EPSPs and explicitly state if a dendritic spike was detected.

4) In the pharmacology studies, it is not clear what kind of criteria the authors used to determine how much drug reached how large of an area. Was drug just applied until there was an effect? What was the point of the lidocaine experiments? It does not seem surprising that blocking either voltage-gated channels or AMPA receptors would largely suppress signals; were these meant as controls for comparing the CGP and AIDA experiments or was there some question as to whether certain effects might actually be insensitive to blockade of Na channel or fast glutamatergic transmission? Also, why were GABA-B receptors chosen (as opposed to GABA-A receptors, or as opposed to other metabotropic receptors)? The experiments require a better rationale.

We added the following paragraph to the Materials and methods section:

“Although it is not possible to confirm the exact concentration of drug applied in vivo, this procedure delivered a volume of solution in the nl range which spreads homogenously throughout the tissue and remained relatively localized (250~500um3) to the injection area. […] An example of drug washout is shown in Figure 2—figure supplement 7.”

We re-wrote the manuscript to make the rational clearer.

We removed the data involving inhibitory synaptic input because this distracted from the major focus (parallel fiber and climbing fiber input). We agree with the reviewer that this aspect requires further experiments since GABA_A, GABA_B and also calcium- and voltage-activated potassium channels in PN dendrites potentially contribute to the hyperpolarizing signals that we see, and we do not yet know how they interact. We are currently exploring this in more detail and will address this in our next manuscript.

5) I think the main discovery is intended to be Figure 6F in which the Ca signal is enhanced when dendritic spikes, presumably from PF activity, follow CF activity, and then Figure 7 makes the case that mGluR1 receptors contribute to the supralinearity. It might be useful to anticipate this link and/or bring up a little more about what is known about mGluR1 signaling in Purkinje cells in both Introduction and Discussion.

We appreciate this comment. We added mGluR1 to the Introduction and Discussion.

Introduction: “These mechanisms involved voltage-gated calcium channels and/or group 1 metabotropic glutamate receptors (mGluR1). How these processes occur during behavior remains to be determined.”

Subsection “Dendritic coincidence detection mechanisms modulate sensory stimulation-evoked dendritic calcium signals in Purkinje neuron dendrites”: “The third mechanism we describe is through mGluR1 activity (Figure 5), which is known to trigger supralinear calcium signals when PF and CF inputs are conjunctively activated, and thereby modify PF-PN synaptic strength (Sarkisov and Wang, 2008; Wang et al., 2000). It was interesting to find that mGluR1 activity can directly influence dendritic spike generation of the CF input and the time-course of the resulting dendritic calcium signal, specifically during evoked DCS events (c.f. Figure 5H and H). Such an interaction between mGluR1 activity and spikelet generation, has previously been postulated (Tempia et al., 2001) and is likely mediated through their interaction with T and P/Q-type VGCCs (Hildebrand et al., 2009; Otsu et al., 2014).”

Reviewer #3:The authors use in vivo two-photon imaging to simultaneously monitor calcium and membrane potential in cerebellar PN dendrites. The sophisticated techniques have been carefully developed by the investigators. The resulting data is compelling, and could ultimately clarify how PN dendrites integrate inputs to modulate postsynaptic activity and synaptic plasticity. The central finding here is that sensory-evoked input drives sub- and supra-threshold signals in PN dendrites. Coincidence of these signals (which is interpreted as originating from PFs and CFs, respectively) produces supralinear calcium responses in PNs. Despite the beauty of the data, and the clear utility of these techniques to investigate the mechanisms of motor learning in the cerebellum, the interpretation of the synaptic source of dendritic signals (particularly PFs) goes beyond the data. However, this central findings are still of great interest to the field.1) Based on their previous work the authors state definitively that DS signals are the result of PF input (Roome and Kuhn, 2018). However, they previously "predicted" that PFs drove DS signals. Subthreshold dendritic depolarizations are similarly inferred to originate from PF input without evidence (Subsection “Sensory stimulation evokes sub- and suprathreshold signals in Purkinje neuron dendrites of awake mice”). While DS and subthreshold signals are very likely to originate from PF input, language throughout the manuscript could easily be amended to reflect the fact that this is a conjecture. Alternatively, new experiments could be used to manipulate granule cell activity (optogenetics, DREADDs.) or selectively inhibit transmission from PFs (for example, release from PFs, but not CFs, is blocked by the group III mGluR agonist L-AP; Hashimoto and Kano, 1998; Kreitzer and Regehr, 2001). Attributing DCS signals to CFs is more clearly justified.

The authors want to thank reviewer #3 for her/his insightful comments.

To address the comments, we performed an additional set of experiments (Figure 1) whereby granule cells were electrically stimulated to evoke PF synaptic input onto PN dendrites (including DS events) and we recorded the resulting dendritic voltage and calcium signals in vivo (see also Figure 4—figure supplement 1). We also revisit the evidence for attributing DS events to PF input first given in (Roome and Kuhn, 2018), showing their blockade by CNQX, and use our DS, DCS sorting algorithm from this study on our paired dendritic/soma recordings to confirm that the DS events have no CF evoked complex spike signal at the soma, and so are evoked by an alternative excitatory synaptic input to the PN dendrites, which leaves only the PFs.

2) While the mechanism of DS signals might be over-interpreted, the mechanisms that motivated pharmacological experiments are barely discussed, leaving the reviewer uncertain as to why these experiments are included. For example, GABAB receptors are noted as present at the PF-PN synapse but their localization and function are not discussed, no prediction is made about the effect of blockade, and these experiments are not discussed outside the Results section. In a manuscript full of excellent data, these experiments should be omitted or explained in greater detail.

We agree with reviewer #3 and re-wrote the manuscript. We removed the data involving inhibitory synaptic input because this distracted from the major focus (parallel fiber and climbing fiber input). The aspect of inhibition requires further experiments since GABA_A, GABA_B and also calcium- and voltage-activated potassium channels in PN dendrites potentially contribute to the hyperpolarizing signals that we see. We do not yet know how they interact – we are currently exploring this in detail.

[Editors' note: further revisions were suggested prior to acceptance, as described below.]

Reviewer 3:1) Subsection “Dendritic spikes triggered by coincident CF and PF synaptic input evoke supralinear calcium signals during sensory stimulation”: Is the point here that under single-DCS conditions, the voltage traces do not show signs of voltage-based amplification, yet the calcium signals still show enhancement? And is the additional source of calcium potentially a source of calcium release or influx that is not voltage-dependent? If so, it would help the reader if you ended the paragraph with an explanatory statement.

Yes, this is correct – although rather than “not voltage-dependent”, we think “not spike-dependent” is more actuate to state at this point, since the enhancement could be through the voltage-dependent (but non-spiking) mechanism described in Figure 3.

We added this to the Results section:

“These differences in the calcium signal time-course are indicative of an additional source of calcium following the evoked DCS, causing a delayed and enhanced calcium peak, while the similarity in spikelet numbers between evoked and non-evoked DCS suggests this additional calcium source is spike-independent. Thus, the enhanced calcium signal is not due to an increase in CF input frequency, or an increase in dendritic spiking, caused by stronger CF input, but though an additional mechanism, that does not require dendritic spikes, such as described in Figure 3, or potentially calcium released from internal stores (Llano et al., 1994; Llano et al., 1991a; Wang et al., 2000).”

2) Subsection “Dendritic spikes triggered by coincident CF and PF synaptic input evoke supralinear calcium signals during sensory stimulation”, supralinearity of calcium signals: Have you tried doing the same analysis but with the voltage signals?

This is an interesting question and something that requires further experiments to explore in more detail. We hope to include this in the next paper.

3) Subsection “Dendritic spikes triggered by coincident CF and PF synaptic input evoke supralinear calcium signals during sensory stimulation”, Figure 4H/I: The timing-dependence curve here shows peak calcium signals at DCS preceding DS by 0-80 ms. If the DCS were a CF-evoked complex spike only, this would seem to be unexpected – it is opposite to timing-dependence measured in brain slices. Since the graph is made in the same style as a similar test in Wang/Denk/Hausser, the discrepancy comes to mind. It could be tied into the next paragraph, where you then explain that the DCS isn't just a complex spike, but has a pre-depolarization. (Can you report near here in the text the duration of the pre-DCS voltage response for comparison?) Also, the DS comes as the culmination of integration of PF input. Again, discussion of the duration of that signal would be helpful. Generally, a critical comparison with the Wang et al. PF-before-CF timing-dependence finding would be useful. I think the two findings are consistent, but it needs explaining.

We added this to the Discussion:

“Here we found the greatest enhancement in calcium signal occurred when the DS followed the DCS by ~ 28 ms, which appears to be in contrast to in-vitro experiments, where the maximal enhancement in calcium signal was detected when PF stimulation preceded CF stimulation (Wang et al., 2000). However, the DS events are due to a culmination of integrated PF input, and so the timing of the DS event does not indicate the onset of PF input, but the moment that the threshold is reached to trigger the DS event. Indeed, we found that on average PF input onset precedes CF input during sensory stimulation by ~ 34 ms, as shown in Figure 2J and Figure 4B, and as indicated by the pre-DCS depolarization in Figure 5C.”

4) Figure 4H: In the 0-80 ms timing window, the relative calcium peak appears to be largest for 2 or 3 spikelets, and smaller for 4 or 5 spikelets. That is unexpected. Or maybe it is just hard to read the graph. Report statistics on that please.

The statistics on this have now been added – the relative calcium peak appears to be independent of the number of DCS spikelets.

5) To address the issue of motion artifacts (reviewer #1 comment 1) the authors suggest that A) the different time-course of voltage and calcium signals argues against motion artifacts underlying these signals, and B) the ability to block both signals with CNQX also suggests synaptic origin. They further state that some analysis techniques were employed to compensate for motion (Subsection “Data analysis”), and mention in the rebuttal that trials were omitted if the mouse made a large movement. Neither of these methods seems to conclusively rule out movement artifacts. This caveat should be mentioned more explicitly in the manuscript.

Yes, we have now included this.

6) In general, the textual revisions have greatly improved the readability and logical flow of the manuscript. However, this sentence in the abstract is still confusing and should be rewritten to better align with their new presentation of the data: "Sensory stimulation evokes post-synaptic potentials that coincide with climbing fiber evoked dendritic complex spikes and parallel fiber evoked dendritic spikes." Where do the spikes come from if not from post-synaptic potentials? Possible solutions would be to replace "coincide with" to "drive", or something like "Sensory stimulation increases the rate of postsynaptic potentials and dendritic calcium spikes evoked by both climbing fiber and parallel fibers." The rewritten statement should also be sure to clarify the "these" in the subsequent sentence.

We changed to this:

“Sensory stimulation increases the rate of post-synaptic potentials and dendritic calcium spikes evoked by climbing fiber and parallel fiber synaptic input.”

7) Since the previous submission, another paper has come out that the authors should mention alongside their treatment of the Gaffield and Christie paper in the Discussion: Roh et al., 2020.

Done – it is an interesting study.